# An EDS1 heterodimer signalling surface enforces timely reprogramming of immunity genes in *Arabidopsis*

Deepak D. Bhandari [1], Dmitry Lapin [1], Barbara Kracher[1], Patrick von Born[1], Jaqueline Bautor[1], Karsten Niefind[2] & Jane E. Parker [1]

Plant intracellular NLR receptors recognise pathogen interference to trigger immunity but how NLRs signal is not known. Enhanced disease susceptibility1 (EDS1) heterodimers are recruited by Toll-interleukin1-receptor domain NLRs (TNLs) to transcriptionally mobilise resistance pathways. By interrogating the *Arabidopsis* EDS1 α-helical EP-domain we identify positively charged residues lining a cavity that are essential for TNL immunity signalling, beyond heterodimer formation. Mutating a single, conserved surface arginine (R493) disables TNL immunity to an oomycete pathogen and to bacteria producing the virulence factor, coronatine. Plants expressing a weakly active EDS1$^{R493A}$ variant have delayed transcriptional reprogramming, with severe consequences for resistance and countering bacterial coronatine repression of early immunity genes. The same EP-domain surface is utilised by a non-TNL receptor RPS2 for bacterial immunity, indicating that the EDS1 EP-domain signals in resistance conferred by different NLR receptor types. These data provide a unique structural insight to early downstream signalling in NLR receptor immunity.

[1] Department of Plant-Microbe Interactions, Max-Planck Institute for Plant Breeding Research, Carl-von-Linne-Weg 10, 50829 Cologne, Germany.
[2] Department of Chemistry, Institute of Biochemistry, University of Cologne, Zuelpicher Strasse 47, 50674 Cologne, Germany. Correspondence and requests for materials should be addressed to J.E.P. (email: parker@mpipz.mpg.de)

Animal and plant innate immunity is governed by surface and intracellular receptors. Mammalian innate immune responses provide an initial barrier against microbial infection, with specific pathogen resistance being taken over by the adaptive immune system. By contrast, plants depend entirely on panels of germ line-encoded receptors[1,2]. Plant recognition of specific virulence factors (effectors) delivered by pathogens to host cells is mediated by intracellular nucleotide-binding domain/ leucine-rich repeat (NLR) receptors in a process called effector-triggered immunity (ETI)[2–4]. NLR-effector recognition leads to induction anti-microbial defence pathways, often accompanied by localised host cell death at infection sites.

Two major plant NLR types are classified by their N-terminal domain architectures: CC-NLRs (or CNLs) have a coiled-coil (CC) domain and TIR-NLRs (TNLs) a Toll-interleukin 1 receptor (TIR) domain, which contribute to NLR activation and signalling[1,4]. A characteristic feature of ETI mediated by the different NLR types is amplification of a similar suite of defence pathways that are mobilised at a lower level by surface pattern-recognition receptors (PRRs) recognising pathogen-associated molecular patterns (PAMPs) in PAMP-triggered immunity (PTI)[5–10]. PTI pathways are often targeted by pathogen effectors to promote infection, and in ETI the transcriptional reestablishment and bolstering of immunity outputs is a major driver of resistance[3,11]. Numerous TFs contribute to ETI governed by CNL and TNL receptors[3,12]. Also, certain NLRs have nuclear functions[3,13–16], suggesting that the path between NLR activation and gene expression reprogramming is in some cases short. However, the mechanisms by which NLRs converge on transcriptional defences are not known.

Another emerging ETI feature is the strong deployment of alternative (parallel) transcriptional branches, enabling the plant to compensate for disabling of particular host resistance sectors[3,11,17–19]. ETI buffering of defence pathways provides robustness against pathogen interference[19]. One important transcriptionally mobilised resistance sector against biotrophic pathogens is controlled by the stress hormone salicylic acid (SA)[20,21]. SA synthesis mediated by the enzyme isochorismate synthase1 (ICS1)[22] is controlled by an ensemble of TFs operating within an intricate phytohormone network[21,23]. SA synthesis and signalling are targeted by pathogens of different classes, often using effectors to boost the SA-antagonising jasmonic acid (JA) hormone system[24–26]. Coronatine (COR) is a potent SA-antagonising virulence molecule produced by *Pseudomonas* bacteria which promotes disease by mimicking plant endogenous bioactive JA-isoleucine (JA-Ile)[26,27]. Like JA-Ile, COR signals by binding to nuclear F-box protein coronatine-insensitive1 (COI1)-jasmonate ZIM-domain (JAZ) coreceptors, which relieves JAZ repression of a basic helix-loop-helix (bHLH) TF, myelocytomatosis oncogene homolog2 (MYC2)[28,29]. MYC2 is a central TF for JA, ethylene and abscisic acid pathways orchestrating numerous stress outputs, including transcriptional dampening of SA accumulation[28].

TNLs recognising different pathogen effectors signal via the nucleocytoplasmic, lipase-like protein enhanced disease susceptibility1 (EDS1), which mediates TNL-activated transcriptional reprogramming, resistance and host cell death[30–37]. *Arabidopsis* EDS1 forms separate heterodimer complexes with its sequence-related partners, phytoalexin deficient4 (PAD4) or senescence-associated gene101 (SAG101) to function in TNL ETI[38–43]. Analysis of the crystal structure of an EDS1-SAG101 heterodimer, and a structural homology-based model of EDS1-PAD4, showed that the partner N-terminal lipase-like (α/β-hydrolase fold) domains provide a non-catalytic scaffold for binding and to promote contacts between the C-terminal α-helical 'EDS1-PAD4' (EP) domains[42,43]. The EP-domains (PFAM:PF18117 [https:// pfam.xfam.org/family/PF18117]) have no significant structural homologies outside the EDS1 family and their role in EDS1 signalling has not been established[43].

*Arabidopsis* EDS1 and SA signalling pathways operate as genetically parallel, mutually reinforcing resistance sectors[8,17–19,44]. We reported that EDS1-PAD4 heterodimers, besides promoting *ICS1* expression and SA accumulation, antagonise COR-stimulated MYC2 pathways in *Arabidopsis* TNL ETI against *Pst* bacteria[19]. PAD4 interacted with MYC2 in planta likely by indirect association. EDS1-PAD4 antagonism of the COR/JA MYC2-branch was nuclear, coincident with or after MYC2 release from COI1-JAZ nuclear complexes, and independent of EDS1-PAD4 promotion of the ICS1/SA-branch[19]. Hence we proposed a two-pronged EDS1 signalling mechanism in ETI for buffering SA immunity against genetic or pathogen interference[19].

Here we interrogate the role of the functionally uncharacterised EDS1 EP-domain in *Arabidopsis* TNL ETI. We identify a positive surface lining a cavity created by the heterodimer, which is essential for pathogen resistance. In TNL immunity against *Pst* bacteria, the EP-domain confers rapid transcriptional reprogramming, which is needed for countering bacterial COR repression of a set of early immunity genes and for disease resistance. We find that the same EDS1 EP-domain surface is recruited by a CNL receptor (RPS2) for resistance against *Pst* bacteria, and in both TNL and CNL bacterial ETI responses, EDS1 signals via three genetically separable resistance sectors.

## Results

**EDS1 EP-domain cavity residues mediate immune signalling.** The *Arabidopsis* EDS1-SAG101 heterodimer crystal structure reveals a cavity formed by the partner EP-domains with nine EDS1 positively charged amino acids (three lysines, five arginines and a histidine) that are conserved across seed plants (Supplementary Fig. 1A, B)[43]. A similar distribution of positive residues was found in a homology-based structural model of the EDS1-PAD4 heterodimer[43]. We selected five solvent accessible EDS1 lysine (K) and two arginine (R) residues that are not part of the heterodimer interface and mutated these individually to alanines (Supplementary Fig. 1A, C). In a yeast 2-hybrid (Y2H) assay, the EDS1 EP-domain amino acid exchanges did not interfere with EDS1-PAD4 interaction compared to the EDS1[LLIF] lipase-like domain mutant which binds PAD4 very weakly (Supplementary Fig. 1D)[43]. Constructs of wild-type EDS1 cDNA (cEDS1) and EDS1 EP-domain variants under the *EDS1* native promoter and tagged with yellow fluorescent protein (YFP) were transformed into the *Arabidopsis* Col *eds1-2* null mutant[6,43]. Primary (T₁) transformants for each construct were then inoculated with the oomycete pathogen *Hyaloperonospora arabidopsidis* (*Hpa*) isolate EMWA1 to test for TNL (*RPP4*) immunity phenotypes, measured against resistant and susceptible controls[45]. Several EP-domain cavity mutants had reduced *RPP4* resistance (Supplementary Fig. 1C). One mutant, EDS1[R493A], located in the centre of the cavity (Fig. 1a) was chosen for in-depth analysis because it displayed similarly high *Hpa* EMWA1 disease susceptibility as *eds1-2*. Consistent with the T₁ TNL immunity phenotypes, two independent homozygous cEDS1[R493A] transgenic lines (R493A#1 and #2) were fully susceptible to *Hpa* isolate CALA2 (Fig. 1b), recognised in Col by *RPP2A* and *RPP2B* TNL genes[46], and the bacterial pathogen *Pst AvrRps4* recognised by the nuclear TNL pair RRS1S-RPS4[31,47] (Fig. 1c). These data show that EDS1[R493A] compromises immunity governed by TNLs against an oomycete and bacterial pathogen, without breaking EDS1 heterodimer formation.

Further examination of the EDS1-SAG101 and EDS1-PAD4 structures revealed a SAG101 and PAD4 EP-domain surface

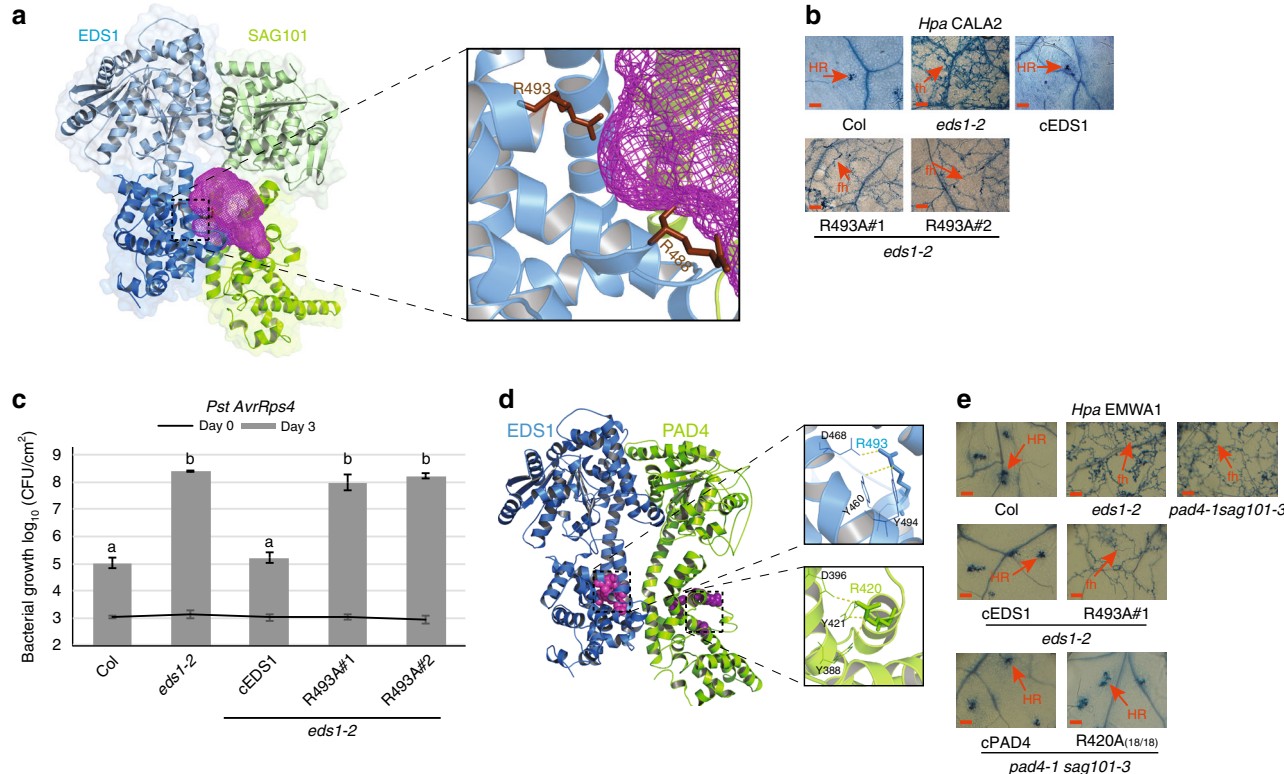

**Fig. 1** Residues lining the EDS1 heterodimer cavity mediate immunity signalling. **a** Crystal structure of EDS1 (blue)–SAG101 (green) showing heterodimer formation chiefly driven by the partner lipase-like domains (light tones) and producing a cavity (magenta mesh) formed by the EP-domains. In the zoom-out, two conserved EDS1 arginine residues lining the cavity are depicted as sticks (brown). **b** *RPP2* resistance phenotypes of 2-week-old control and transgenic lines expressing YFP-cEDS1 or R493A. *Hpa* EMWA1 infected leaves were stained with trypan blue at 5 dpi. Scale bar represents 100 μm. Images are representative of 24 leaves from two independent experiments. HR, hypersensitive response; fh, pathogen free hyphae. **c**. Four-week old *Arabidopsis* plants of the indicated genotypes were infiltrated with *Pst AvrRps4* ($OD_{600}$–0.0005) and bacterial titres determined at 0 and 3 dpi. Bars represent mean of four biological replicates ± SE. Differences between genotypes were determined using ANOVA (Tukey's HSD, $p < 0.005$). Similar results were obtained in three independent experiments. **d** Homology model of EDS1 (blue)–PAD4 (green). Conserved residues in the EP-domains of EDS1 (magenta) and PAD4 (purple) are represented as spheres. EDS1 residues line the heterodimer cavity while PAD4 residues are not part of the cavity. A zoom-out of the sphere-represented residues shows ionic and hydrogen bonds formed by $EDS1^{R493}$ and equivalent arginine in $PAD4^{R420}$ with neighbouring residues. **e** *RPP4* resistance phenotypes of 2-week-old control and homozygous transgenic lines expressing wild-type EDS1, PAD4 and mutated arginine variants. *Hpa* EMWA1 infected leaves were stained with trypan blue at 5 dpi. Scale bar represents 100 μm. The PAD4 R420A image is representative of 18 independent transgenic (T1) plants. HR hypersensitive response, fh pathogen free hyphae

resembling the $EDS1^{R493A}$ patch but facing away from the cavity (Fig. 1d). We mutated PAD4 arginine 420 ($PAD4^{R420A}$) which is aligned at the sequence level and predicted to form similar interactions with neighbouring residues as $EDS1^{R493}$, but on the external PAD4 surface (Fig. 1d). YFP-tagged wild-type cPAD4 and $PAD4^{R420A}$ under the *PAD4* native promoter were transformed into a Col *pad4-1 sag101-3* mutant, in which loss of *SAG101* is compensated for by *PAD4*[41,43]. In *RPP4* (*Hpa* EMWA1) infection assays of T1 transgenic lines, $PAD4^{R420A}$ was as resistant as cPAD4 and Col, whereas the EDS1 R493A#1, *eds1-2* and *pad4-1 sag101-3* plants were susceptible (Fig. 1e). These data suggest the location of positively charged residues such as $EDS1^{R493}$ within the EP-domain cavity is critical for TNL immunity.

We tested whether $EDS1^{R493A}$ retains interaction with PAD4 in plant cells by performing transient expression and immuno-precipitation (IP) assays in *eds1-2 pad4-1 sag101-3* protoplasts[19]. $EDS1^{R493A}$ fused to a FLAG tag interacted with PAD4-YFP as strongly as wild-type EDS1-FLAG, whereas a non-interacting $EDS1^{LLIF}$-FLAG variant did not bind PAD4-YFP (Supplementary Fig. 2A). YFP-tagged $cEDS1^{R493A}$ protein in R493A#1 and R493A#2 had a similar nucleocytoplasmic distribution as YFP-cEDS1 at 24 hpi with *Pst AvrRps4* (Supplementary Fig. 2B),

indicating that loss of TNL immunity is not due to failed EDS1 nuclear accumulation[35,36]. Confocal microscopy imaging of multiple samples showed that the YFP-$cEDS1^{R493A}$ nucleocytoplasmic fluorescence signal was lower than for YFP-cEDS1 (Supplementary Fig. 2B). Also, R493A lines #1 and #2 accumulated less total EDS1 protein compared to cEDS1 (Supplementary Fig. 2C). We tested whether R493A susceptibility could be due to low protein accumulation by selecting independent homozygous transgenic lines (#1 and #2) for two other YFP-cEDS1 EP-domain mutants: K487A and K387A, which had displayed full *RPP4* resistance in the T1 seedling assays (Supplementary Fig. 1C) and similar or lower basal EDS1 protein accumulation than R493A lines #1 and #2 (Supplementary Fig. 2C). The K487A and K387A lines were fully resistant to *Pst AvrRps4* (Supplementary Fig. 2D) and *Hpa* EMWA1 (Supplementary Fig. 2E). Additionally, transgenic *eds1-2* lines expressing genomic EDS1-YFP or $EDS1^{R493A}$-YFP (denoted gEDS1 and gR493A) were selected because gEDS1 is generally more highly expressed than cEDS1[36]. Although gEDS1 and gR493A accumulated to similar levels as cEDS1 in mock-treated tissues and after *Pst AvrRps4* infection (Supplementary Fig. 3A), the gR493A transgenic lines were fully susceptible to *Pst AvrRps4* (Supplementary Fig. 3B). The cR493A and gR493A lines were

similarly defective in *Pst AvrRps4* induced expression of *EDS1*-dependent defence marker genes[6,35] *PAD4*, *PBS3*, *ICS1* and *FMO1*, and in repression of *MYB34*, measured by qRT-PCR at 8 hpi (Supplementary Fig. 3C). Also, cR493A and gR493A lines displayed full susceptibility to *Hpa* EMWA1 infection (Supplementary Fig. 3D). These data show that impaired TNL *RRS1S RPS4* and *RPP4* resistance in the R493A lines is not a consequence of low EDS1[R493A] protein accumulation. We concluded that EDS1[R493] lining the EP-domain cavity confers an important signalling property on the EDS1-PAD4 heterodimer.

**EDS1[R493A] delays TNL transcriptional reprogramming**. TNL/EDS1 bacterial immunity divides into two mutually reinforcing resistance branches: one requiring SA synthesised by EDS1-induced *ICS1*, the other independent of SA and involving EDS1 antagonism of MYC2[17,19]. We determined whether SA accumulation is altered in R493A by measuring free (active) SA in leaves of Col, *eds1-2* and cEDS1 and R493A plants at 0, 8 and 24 hpi with *Pst AvrRps4*. At 8 hpi, SA levels in R493A lines #1 and #2 remained low, resembling the *eds1-2* null mutant (Fig. 2a). At 24 hpi, the R493A lines but not *eds1-2*, recovered SA accumulation to similar levels as cEDS1 and Col (Fig. 2a). Low SA accumulation at 8 hpi correlated with reduced expression of the SA-marker gene *PR1* (*pathogenesis related1*) in R493A compared to cEDS1 and Col at 24 hpi (Fig. 2b). These data show that EDS1[R493A] is not a complete loss-of-function mutation but instead slow to mobilise the SA branch of TNL immunity against *Pst AvrRps4* bacteria. Therefore, one function of the EDS1 EP-domain, which is compromised in R493A, is to promote SA immunity.

We interrogated the EDS1[R493A] defect in TNL (*RRS1S RPS4*) transcriptional reprogramming further using RNA-seq analysis. Four-week-old Col, *eds1-2*, cEDS1 and R493A#1 plants were infiltrated with *Pst AvrRps4* and three independent biological replicates for each line analysed at 0, 8 and 24 hpi (see Methods). At 8 hpi there were only 12 DEGs between R493A and *eds1-2* (Fig. 2c (pink boxes) and Supplementary Fig. 4A; Supplementary Table 1). These included *EDS1* and *PBS3*. Therefore, at the level of *RRS1S RPS4*-triggered SA accumulation and transcriptional reprogramming, R493A behaves like the *eds1-2* null mutant at 8 hpi (Fig. 2a, c and Supplementary Fig. 4A). At 24 hpi, R493A had a markedly different expression profile to *eds1-2* (Fig. 2c and Supplementary Fig. 4A) with 2053 DEG. R493A did not fully recover at 24 hpi as there were 576 DEG between R493A and cEDS1 compared with 5993 DEG between *eds1-2* and cEDS1 (Fig. 2c and Supplementary Fig. 4A). These gene expression profiles show that EDS1[R493A] fails to mobilise TNL transcriptional reprogramming at 8 hpi but recovers substantially at 24 hpi. Based on the R493A bacterial disease susceptibility phenotype (Fig. 1c and Supplementary Fig. 2D), we concluded that recovery of gene expression in R493A lines at 24 hpi is too late to halt pathogen growth. Hence, amino acid R493 in the EDS1 EP-domain lining the heterodimer cavity is critical for timely TNL transcriptional defence reprogramming in response to *Pst AvrRps4*.

**EDS1[R493A] fails to antagonise bacterial COR-stimulated MYC2 in ETI**. One function of the EDS1-PAD4 heterodimer in *RRS1S RPS4* ETI is to inhibit COR stimulation of MYC2-regulated JA pathway genes independently of *ICS1*[19]. Because R493A plants displayed a general delay in gene expression reprogramming (Fig. 2c) and in accumulation of SA (Fig. 2a), we tested whether EDS1[R493A] is defective in antagonising COR-promoted bacterial infection. For this, *Pst AvrRps4* or the COR-deficient *Pst Δcor AvrRps4* strain were infiltrated into Col, *eds1-2*, cEDS1 and

R493A leaves and bacterial growth measured at 3 dpi. As expected, Col and cEDS1 were equally resistant to *Pst AvrRps4* and *Pst Δcor AvrRps4*, reflecting EDS1 antagonism of COR-stimulated bacterial growth in *RRS1S RPS4* ETI (Fig. 3a). The *eds1-2* mutant was susceptible to both *Pst AvrRps4* strains but supported 1.5-log lower *Pst Δcor AvrRps4* growth compared to *Pst AvrRps4*, indicating both COR-stimulated and COR-independent bacterial growth promotion in the absence of *EDS1* (Fig. 3a)[19]. Notably, while R493A lines displayed high susceptibility to *Pst AvrRps4*, they were as resistant as Col and cEDS1 to *Pst Δcor AvrRps4* (Fig. 3a). Therefore, loss of R493A resistance to *Pst AvrRps4* is due to its failure to antagonise bacterial COR. We checked whether COR activity in R493A signals via MYC2 by crossing cEDS1 and R493A#1 into an *eds1-2 myc2-3* mutant background. Whereas R493A (in *eds1-2*) was as susceptible as *eds1-2*, R493A *eds1-2 myc2-3* was as resistant as Col and cEDS1 (Fig. 3b), indicating that EDS1[R493A] recovers resistance to *Pst AvrRps4* when *MYC2* is mutated. Hence, EDS1[R493A] in R493A#1 has a fully restored resistance function in the absence of COR or *MYC2*. These data expose a defect of EDS1[R493A] in counteracting COR-dependent, MYC2-promoted bacterial growth in TNL ETI. With respect to *Pst AvrRps4*, therefore, EDS1[R493A] loss of resistance is conditional on COR stimulation of the MYC2 JA signalling branch.

As EDS1[R493A] compromised TNL (*RPP4* and *RPP2A, B*) immunity to *Hpa* (Fig. 1b, e and Supplementary Fig. 2E, 3D), we tested whether there is also recovery of *RPP4* resistance to *Hpa* EMWA1 in the *eds1-2 myc2-3* transgenic lines. Col, *myc2-3*, cEDS1 in *eds1-2* and cEDS1 in *eds1-2 myc2-3* expressed full *RPP4* resistance after quantifying EMWA1 sporulation on leaves (Fig. 3c). There were similar high levels of EMWA1 sporulation in *eds1-2*, *eds1-2 myc2-3*, R493A *eds1-2* and R493A *eds1-2 myc2-3* (Fig. 3c). These data show that increased susceptibility to *Hpa* EMWA1 in R493A is not via *MYC2*. We concluded that the EDS1 EP-domain and associated heterodimer cavity have broader functions in TNL immunity than antagonising MYC2.

**TNL/EDS1 signalling delay is exploited by bacterial COR**. Because COR-activated MYC2 represses SA accumulation[19,26,27] and EDS1[R493A] delays SA accumulation at 8 hpi with *Pst AvrRps4* (Fig. 2a), we tested whether R493A resistance to *Pst Δcor AvrRps4* (Fig. 3a) is due to restored SA. Col and cEDS1 accumulated similar free SA levels at 8 hpi with *Pst AvrRps4* and higher SA in response to *Pst Δcor AvrRps4*, consistent with COR dampening SA accumulation (Fig. 4a). The *eds1-2* null mutant failed to accumulate SA in response to either strain (Fig. 4a), fitting with EDS1 promotion of *ICS1* expression and SA independently of its suppression of MYC2 in TNL (*RRS1S RPS4*) immunity[19]. Strikingly, R493A lines failed to accumulate SA in response to *Pst AvrRps4* and *Pst Δcor AvrRps4* (Fig. 4a). Therefore, delayed SA accumulation in R493A is not caused by a failure to antagonise bacterial COR. These data suggest that antagonism of COR/MYC2 signalling and promotion of SA accumulation are distinct properties of the EDS1-PAD4 heterodimer EP-domain, and more precisely EDS1[R493] lining the EP cavity, in TNL ETI to *Pst AvrRps4*.

Next we explored which gene expression sectors affected by bacterial COR might explain the compromised TNL ETI in R493A by performing RNA-seq analysis at 8 and 24 hpi with *Pst Δcor AvrRps4*. To compare between *Pst AvrRps4* and *Pst Δcor AvrRps4* RNA-seq experiments, gene expression data for all lines was normalised against the respective Col control within each treatment (see Methods). A multi-dimensional scaling (MDS) plot shows that cEDS1 and Col expression profiles clustered together at 8 and 24 hpi for both treatments (Fig. 4b). At 8 hpi,

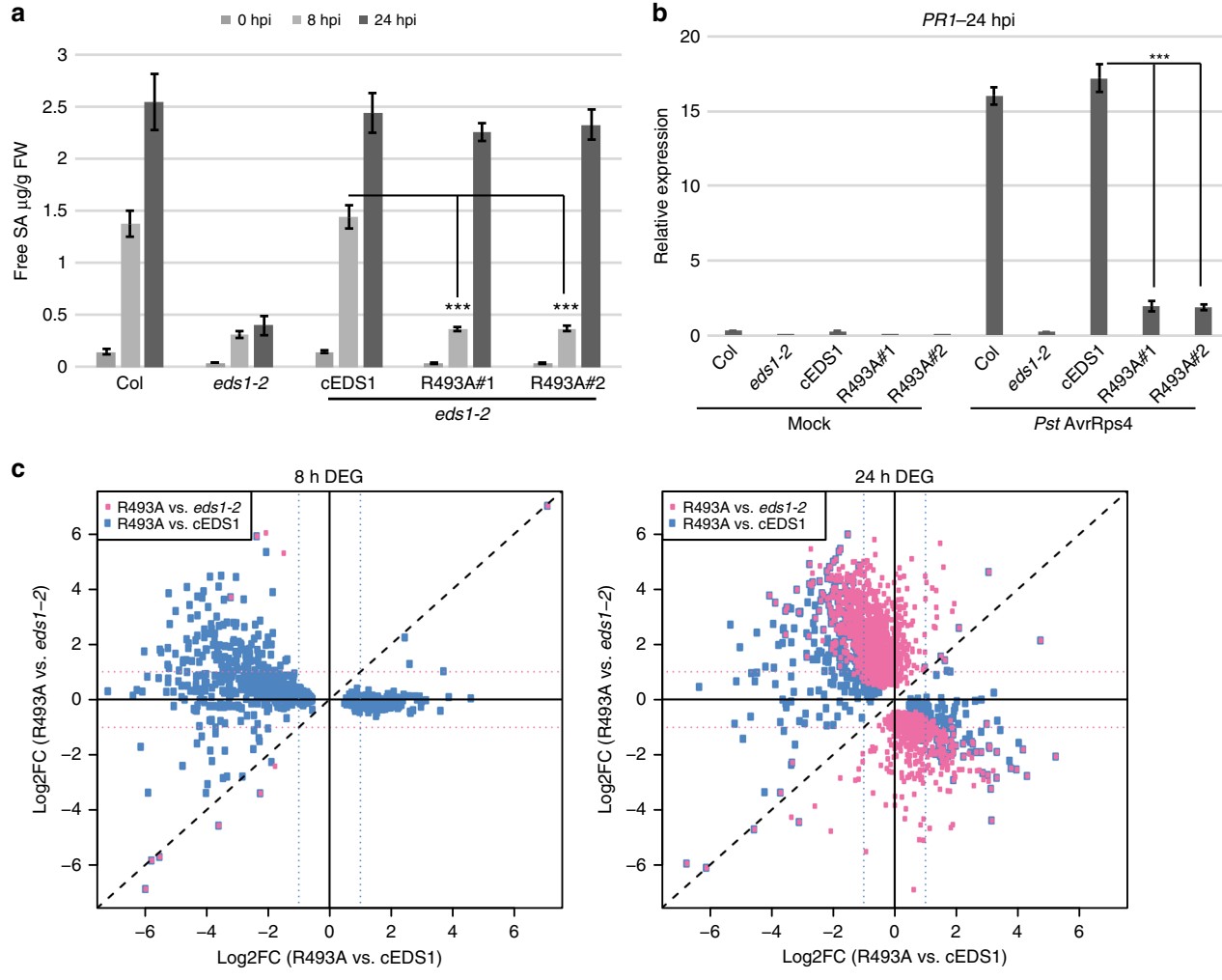

**Fig. 2** EDS1$^{R493A}$ delays TNL transcriptional reprogramming. **a** Four-week-old plants were infiltrated with *Pst AvrRps4* (OD$_{600}$–0.005) and free SA was quantified at 0, 8 and 24 hpi. Bars represent means ± SE of four biological replicates. Differences between genotypes were analysed using t-test (Bonferroni corrected, $p < 0.05$). Similar results were obtained in two independent experiments. **b** Four-week-old plants were infiltrated with 10 mM MgCl$_2$ (mock) or *Pst AvrRps4* (OD$_{600}$–0.005), and leaf samples collected at 24 hpi. *PR1* transcripts were measured using qRT-PCR and normalised to *GapDH*. Bars represent means ± SE of three biological replicates. Differences between genotypes were analysed using *t*-test (Bonferroni corrected, $p < 0.05$). **c** A 2-D scatter plot comparing R493A#1 vs. *eds1-2* and R493A#1 vs. cEDS1 at 8 and 24 hpi with *Pst AvrRps4*. Plots depict differentially expressed genes (DEG) between R493A vs. *eds1-2* (pink dots) and R493A vs. cEDS1 (blue dots). DEG represented were filtered with a |log$_2$ FC| ≥ 1, FDR ≤ 0.05

R493A clustered away from cEDS1 and *eds1-2* with *Pst Δcor AvrRps4* but close to *eds1-2* with *Pst AvrRps4* (Fig. 4b, coloured circles). At 24 hpi, the R493A transcriptome was similar to cEDS1 with *Pst Δcor AvrRps4* (filled triangles) but distinct from cEDS1 with *Pst AvrRps4* (filled squares) (Fig. 4b). This analysis shows that EDS1$^{R493A}$ causes a general delay in TNL gene expression reprogramming which is exacerbated by bacterial COR. The results underscore importance of the EDS1 EP-domain for rapid transcriptional mobilisation of multiple pathways besides blocking bacterial COR actions in TNL ETI. We compared our transcriptome data at 8 hpi with publicly available SA-responsive and JA-responsive transcriptomes[8,48]. This analysis showed that R493A exhibits lower expression of genes regulated by *ICS1* compared to cEDS1 regardless of bacterial COR status (Supplementary Fig. 4B). In contrast, genes that are repressed by JA are reduced in R493A with *Pst AvrRps4* but not with *Pst Δcor AvrRps4* (Supplementary Fig. 4B). Thus, the R493A transcriptome reflects both slow mobilisation of SA pathways and defective antagonism of COR/MYC2-regulated JA pathways.

We searched for DEG between *Pst AvrRps4* and *Pst Δcor AvrRps4* treatments in the different lines at 8 and 24 hpi. After

hierarchical clustering, a derived expression heatmap emphasised the extent of delay in R493A at 8 hpi compared to cEDS1, in the presence and absence of bacterial COR (Fig. 4c). Only 75 (8 hpi) and 3 (24 hpi) DEG spread across different clusters were found for cEDS1, consistent with EDS1 effectively antagonising COR in TNL ETI. While R493A was slow in ETI transcriptional reprogramming against both *Pst AvrRps4* and *Pst Δcor AvrRps4* (Fig. 4b), there remained 1009 (8 hpi) and 210 (24 hpi) DEG between these treatments in R493A (Supplementary Table 2), indicating that the transcriptional delay in R493A is compounded by COR. Because R493A is susceptible to *Pst AvrRps4* but resistant to *Pst Δcor AvrRps4* infection (Fig. 3a), we concluded that EDS1 early interference with COR-dependent expression changes (before or at 8 hpi) is important for inhibiting bacterial growth in the TNL immune response.

**COR represses a set of immunity-related genes in R493A.** One expression cluster (#17) stood out in the above analysis because it contains EDS1-dependent DEG at 8 hpi that are more highly expressed in R493A with *Pst Δcor AvrRps4* compared to *Pst AvrRps4* (Fig. 4c, Supplementary Data 1), suggesting there is COR

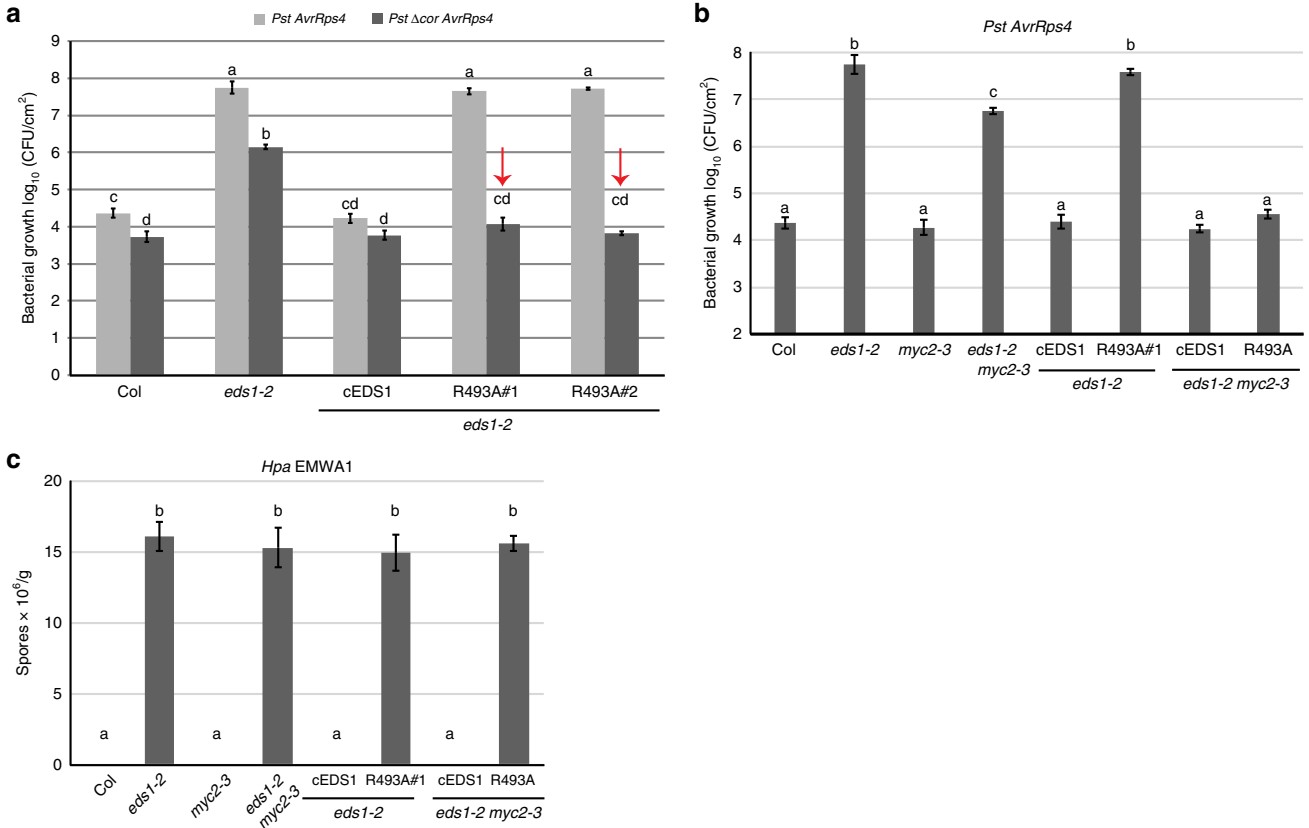

**Fig. 3** EDS1$^{R493A}$ fails to antagonise bacterial COR-stimulated MYC2 in TNL immunity. **a** Four-week-old *Arabidopsis* plants of the indicated genotypes were infiltrated with *Pst AvrRps4* or *Pst ΔCor AvrRps4* (OD$_{600}$–0.0005). Bacterial titres were determined at 0 and 3 dpi. No significant difference was observed between lines and treatments at 0 dpi. Bars represent mean of four biological replicates ± SE. Differences between genotypes were analysed using ANOVA (Tukey's HSD, *p*-value < 0.005). Similar results were obtained in three independent experiments. **b** Four-week-old *Arabidopsis* plants of the indicated genotypes were infiltrated with *Pst AvrRps4* (OD$_{600}$– 0.0005). Bacterial titres were determined at 0 and 3 dpi. No significant differences were observed at 0 dpi. Bars represent mean of four biological replicates ± SE. Differences between genotypes were analysed using ANOVA (Tukey's HSD, *p*-value < 0.005). Similar results were obtained in three independent experiments. **c** The indicated genotypes were infected with *Hpa* EMWA1 and oomycete sporulation was quantified at 5 dpi. Data from three independent experiments were combined and differences between genotypes analysed using ANOVA (Tukey's HSD, *p* < 0.005)

repression of a set of genes in R493A but not in cEDS1 (which counters COR effects) in TNL immunity. Cluster #17 comprises 383 genes associated with Gene Ontology (GO) terms phosphorylation (37/376 eg. *MPK3*, *MPK2*), cell death (18/376 eg. *RPS2*, *NPR1*, *CPR5*) and defence responses (73/376 eg. *RPS4*, *RPP4*, *ADR1-L2*) (Supplementary Data 2). Notable members of cluster #17 are functionally defined *NLR* (*TNL* and *CNL*) and *WRKY* family TF genes (Fig. 4c, Supplementary Data 1).

We found a significant overlap between cluster #17 genes and genes regulated by the SA analogue Benzothiadiazole (BTH)[25] or JA[48] (Fig. 4d; Supplementary Data 3). This comparison suggests that early in bacterial infection, TNL/EDS1 signalling protects a large number of SA-promoted/JA-repressed (169/383) genes but also a set of SA/JA-unrelated immunity genes (82/383) from bacterial COR repression. One third (118/383) cluster #17 genes overlap with a set of ETI-associated genes extracted from transcriptomic analyses of the *Arabidopsis RPS2* (CNL) response to AvrRpt2 expressed *in planta* or delivered by *Pst* bacteria[9,49,50] (Supplementary Fig. 4C). Therefore, cluster #17 likely contains some new ETI-related genes. We did not detect significant enrichment of particular *cis*-regulatory elements in the promoters of cluster #17 genes (by MEME, http://meme-suite.org), suggesting that these genes are controlled by multiple TFs.

In summary, comparing cEDS1, *eds1-2* and R493A transcriptomes between *Pst AvrRps4* and *Pst Δcor AvrRps4* treatments

uncovered a set of TNL/EDS1-controlled genes (cluster #17) whose repression in response to bacterial COR at 8 hpi is inadequately blocked by EDS1$^{R493A}$. Therefore, R493A highlights a role of the EDS1 heterodimer EP-domain in countering bacterial COR repression of host immunity genes within the first 8 h of TNL immunity.

**A positive charge at EDS1$^{R493}$ is essential for TNL immunity**. We have shown that mutations of positively charged residues in the EDS1 EP cavity impair TNL immunity (Supplementary Fig. 1C and Fig. 1b, c). To test the importance of the charge at R493, we generated positively (lysine, cEDS1$^{R493K}$) and negatively (glutamate, cEDS1$^{R493E}$) charged variants of EDS1$^{R493}$. Like EDS1$^{R493A}$, YFP-tagged EDS1$^{R493K}$ and EDS1$^{R493E}$ displayed wild-type nucleocytoplasmic localisation in transient expression assays in *N. benthamiana* (Supplementary Fig. 5A). The FLAG-tagged EDS1$^{R493}$ variants interacted with PAD4-YFP in IP experiments (Supplementary Fig. 5B), consistent with the charge at the EP cavity not affecting heterodimer formation. Additionally, the EDS1$^{R493}$ mutations did not alter interactions between EDS1/PAD4-YFP complexes and StrepII-HA(SH)-tagged MYC2 in IPs of transiently expressed proteins (Supplementary Fig. 5C), showing that R493 does not affect EDS1/PAD4 association with MYC2. In these assays, EDS1 but not EDS1$^{LLIF}$ (which fails to bind PAD4 stably) decreased PAD4–MYC2 IP signals

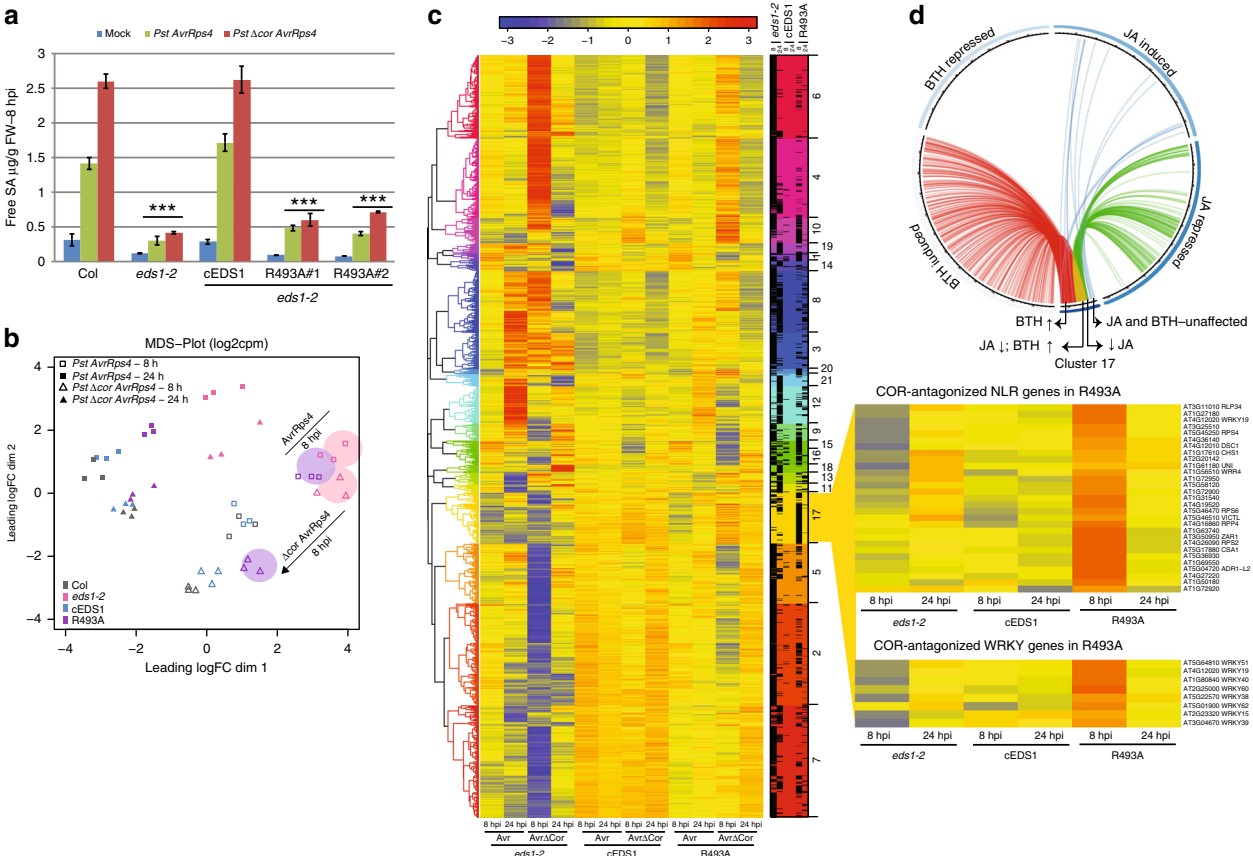

**Fig. 4** Delayed immune signalling in R493A mutant plants is independent of COR. **a** Four-week-old plants were infiltrated with 10 mM MgCl2 (mock), *Pst AvrRps4* or *Pst ΔCor AvrRps4* and free SA was quantified at 8 hpi. Bars represent means ± SE of three biological replicates. Difference between genotypes were analysed using t-test (Bonferroni corrected, $p < 0.05$). Similar results were observed in three independent experiments. **b** A multi-dimension scaling (MDS) plot of differentially expressed genes showing R493A transcriptional changes at 8 (open symbols) and 24 hpi (closed symbols). Encircled samples of *eds1-2* (pink) and R493A (purple) highlight transcriptional trends with and without bacterial COR in *Pst AvrRps4*-triggered immunity. **c** A heatmap depicting DEG at 8 and 24 hpi normalised to Col ($p < 0.05$) after hierarchical clustering. Samples were harvested at 8 and 24 hpi with *Pst AvrRps4* (Avr) and *Pst ΔCor AvrRps4* (ΔCor). Cluster #17 contains genes that are upregulated at 8 hpi with *Pst ΔCor AvrRps4* but not *Pst AvrRps4* in R493A only. Expansion (right) highlights a subset of *NLR* and *WRKY* genes in cluster #17 involved in immunity whose expression is antagonised by COR in R493A (see Supplementary Table 3). **d** Circos plot showing overlap of 383 genes in cluster #17 with genes regulated by JA and Benzothiadiazole (BTH, an analogue of SA) from other studies (Hickmann et al.,2017; Yang et al.,2017). Genes differentially expressed in cluster #17 and other datasets are marked by connecting lines. Genes repressed by JA and induced by BTH (red and green lines converging to yellow, 113); genes repressed by JA and not expressed by BTH (green lines, 19); genes induced by BTH and not expressed in JA dataset (red lines, 169)

(Supplementary Fig. 5C), consistent with earlier evidence that PAD4–MYC2 interaction is competed by EDS1 in IPs and that PAD4–MYC2 binding alone does not explain antagonism of MYC2 in TNL (*RRS1S RPS4*) ETI[19].

Two independent homozygous transgenic lines (#1 and #2) were selected in *Arabidopsis eds1-2* for cEDS1[R493K] and cEDS1[R493E]. In these lines, YFP-tagged EDS1[R493K] protein accumulated to similar or slightly higher levels than EDS1[R493A] whereas EDS1[R493E] accumulation was lower (Supplementary Fig. 6A, B). There was increased accumulation of all EDS1 forms in response to *Pst AvrRps4* at 24 hpi. In *RRS1S RPS4* resistance to *Pst AvrRps4* (Supplementary Fig. 6A, B) and *RPP4* resistance to *Hpa* EMWA1 (Fig. 5a), the R493E lines were as susceptible as *eds1-2* and R493 A, whereas R493K and cEDS1 lines were as resistant as Col. Therefore, a positive charge at EDS1 amino acid 493 rather than arginine *per se* is required for TNL immunity.

Because R493A susceptibility to *Pst AvrRps4* was conditional on COR (Fig. 3a), we tested TNL (*RRS1S RPS4*) responses of R493K and R493E to *Pst AvrRps4* and *Pst Δcor AvrRps4*. Measured against Col, cEDS1 and *eds1-2*, R493K was fully resistant and R493E fully susceptible to both *Pst AvrRps4* strains,

whereas R493A#1 was susceptible only to *Pst AvrRps4* (Fig. 5b). Free SA accumulation in the R493K and R493E lines at 8 and 24 hpi with *Pst AvrRps4* or *Pst Δcor AvrRps4* mirrored, respectively, cEDS1 and *eds1-2* (Fig. 5c).

We next measured the expression of marker genes (*EDS1*, *PAD4* and *ICS1*) for the EDS1/PAD4-induced SA immunity branch at 8 hpi with *Pst AvrRps4* and *Pst Δcor AvrRps4* in the R493K and R493E lines compared to Col-0, *eds1-2*, cEDS1 and R493A#1 (Supplementary Fig. 6C). The disease resistance, SA accumulation and gene expression phenotypes show that R493K phenocopies wild-type *EDS1*, and R493E the *eds1-2* null mutant. We performed qRT-PCR analysis of MYC2-branch JA response marker genes *SA methyl transferase 1* (*BSMT1*), *JAZ10* and *Vegetative Storage Protein 1* (*VSP1*) in the R493 variants at 24 hpi with *Pst AvrRps4*[19]. Here, R493K behaved like wild-type cEDS1 and R493E like *eds1-2* (Supplementary Fig. 6D). By contrast, R493A repressed *BSMT1*, *JAZ10* and *VSP1* MYC2-branch genes nearly as strongly as cEDS1 (or Col) (Supplementary Fig. 6D). Hence, EDS1[R493A] antagonism of the COR-stimulated MYC2-branch at a late time point (24 hpi) is insufficient for restricting *Pst AvrRps4* infection in ETI (Fig. 3a). We concluded that a

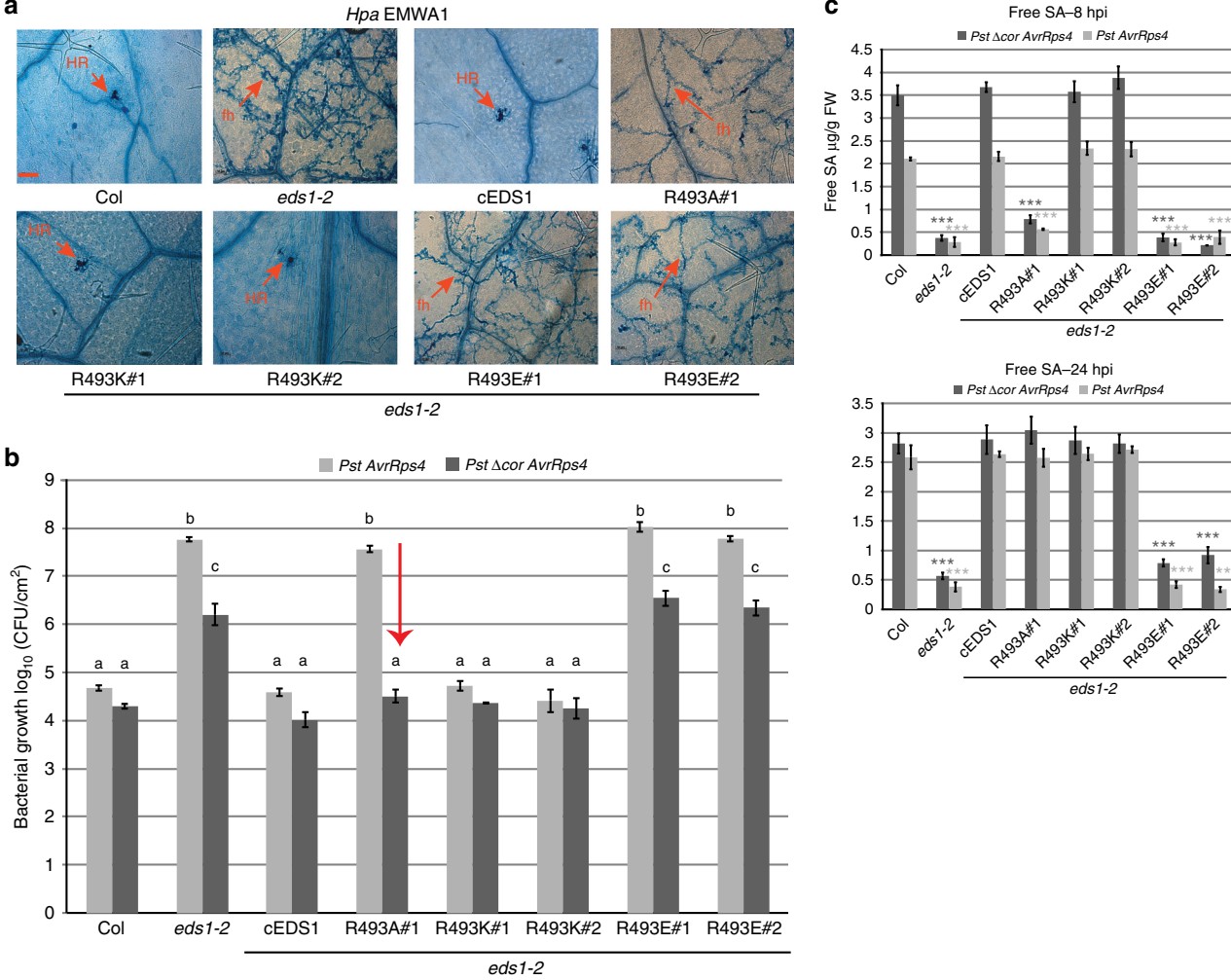

**Fig. 5 a** positive charge at EDS1$^{R493}$ is essential for TNL immunity. *RPP4* resistance phenotypes of 2-week-old control and *Arabidopsis* transgenic lines expressing cEDS1 and R493 mutants, as indicated. *Hpa* EMWA1 infected leaves were stained with trypan blue at 5 dpi. Scale bar represents 100 μm. Each image is representative of >18 leaves from two independent experiments. HR, hypersensitive response; fh, pathogen free hyphae. **b** Four-week-old *Arabidopsis* plants of the indicated genotypes were infiltrated with *Pst AvrRps4* and *Pst ΔCor AvrRps4* (OD$_{600}$–0.0005). Bacterial titres were determined at 3 dpi. Bars represent means of three biological replicates ± SE. Differences between genotypes were analysed using ANOVA (Tukey's HSD, $p < 0.05$). Similar results were obtained in three independent experiments. **c** Four-week-old plants were infiltrated with *Pst AvrRps4* or *Pst ΔCor AvrRps4* and free SA was quantified at 8 and 24 hpi. Bars represent means ± SE of three biological replicates. Differences between genotypes within treatments were analysed using Student's *t*-test (Bonferroni corrected, ***$p < 0.05$) relative to Col. Similar results were obtained in two independent experiments

positive charge at EDS1$^{R493}$ is critical for timely defence gene expression changes and for effective *RRS1S RPS4*-triggered immunity.

**EDS1$^{R493A}$ weak function is responsive to TNL activation.** EDS1/PAD4 complexes confer post-infection basal immunity to virulent *Pst* DC3000 in the absence of TNL-effector recognition[39,40,42,43]. We therefore tested whether EDS1$^{R493A}$ restriction of *Pst* DC3000 growth is conditional on bacterial COR by infiltrating leaves of different EDS1$^{R493}$ variants with *Pst* DC3000 or *Pst Δcor*. Col, cEDS1 and R493K expressed similar resistance to each bacterial strain, with ~10-fold higher *Pst* DC3000 growth than *Pst Δcor* at 3 dpi (Fig. 6a). The *eds1-2*, R493A and R493E lines were similarly susceptible to each strain, with ~100-fold higher *Pst* DC3000 growth compared to *Pst Δcor* (Fig. 6a). Therefore, R493A does not recover basal resistance to *Pst Δcor* in contrast to the recovered TNL resistance to *Pst Δcor AvrRps4* (Fig. 5b). These data show that a positive charge at

EDS1$^{R493}$ is essential for both TNL ETI and basal immunity against *Pst* bacteria, and that EDS1$^{R493A}$ is partially equipped for TNL ETI but unequipped for basal immunity regardless of bacterial COR repressive effects.

**CNL receptor RPS2 signals via the EDS1 EP-domain in ETI.** The *ICS1*/SA pathway partially compensates for *eds1* loss-of-function in TNL ETI mediated by *RRS1S RPS4* and fully for *eds1* in CNL ETI mediated by *RPS2* or *HRT* (HR to turnip crinkle virus)[17,44]. We removed the *ICS1*/SA sector to expose an EDS1-only resistance function by crossing cEDS1 and R493A#1 transgenic lines into an *eds1 ics1* (*eds1-2 sid2-1*) mutant background. Growth of *Pst Δcor AvrRps4* and *Pst AvrRpt2* bacteria was then quantified as a measure, respectively, of TNL (*RRS1S RPS4*) and CNL (*RPS2*) immunity. As expected, cEDS1 and R493A (in *eds1-2*) conferred full TNL immunity to *Pst Δcor AvrRps4*, as did cEDS1 in *eds1-2 sid2-1* (Fig. 6b), consistent with EDS1 compensating for loss of *ICS1*-generated SA in TNL ETI[17]. By

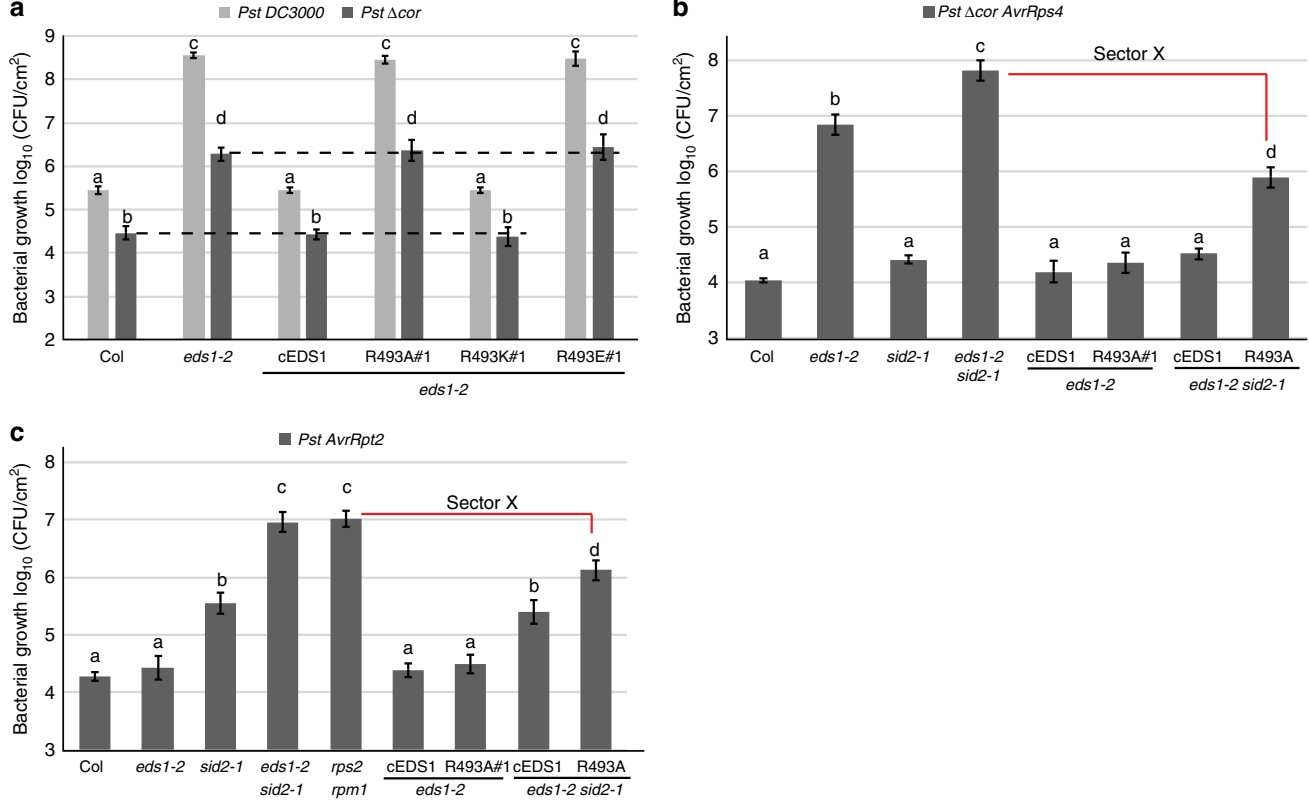

**Fig. 6** EDS1[R493] signals in TNL and CNL (*RPS2*) immunity. **a** Four-week-old *Arabidopsis* plants of the indicated genotypes were infiltrated with *Pst* DC3000 and *Pst* ΔCor (OD$_{600}$–0.0005). Bacterial titres were determined at 0 and 3 dpi. No significant difference was observed between lines and treatments at 0 dpi. Bars represent mean of four biological replicates ± SE. Differences between genotypes were analysed using ANOVA (Tukey's HSD, $p < 0.005$). Similar results were obtained in three independent experiments. **b** Four-week-old *Arabidopsis* plants of the indicated genotypes were infiltrated with *Pst* ΔCor *AvrRps4* (OD$_{600}$–0.0005). Bacterial titres were determined at 0 and 3 dpi. No significant difference was observed at 0 dpi. Bars represent mean of four biological replicates ± SE. Differences between genotypes were analysed using ANOVA (Tukey's HSD, $p < 0.005$). Similar results were obtained in three independent experiments. **c** Four-week-old *Arabidopsis* plants of the indicated genotypes were infiltrated with *Pst* *AvrRpt2* (OD$_{600}$–0.0005). Bacterial titres were determined at 0 and 3 dpi. No significant difference was observed at 0 dpi. Bars represent mean of four biological replicates ± SE. Differences between genotypes were analysed using ANOVA (Tukey's HSD, $p < 0.005$). Similar results were obtained in three independent experiments

contrast, *Pst* Δ*cor AvrRps4* growth in R493A *eds1-2 sid2-1* was intermediate between cEDS1 *eds1-2 sid2-1* and *eds1-2 sid2-1* (Fig. 6b). Therefore, *ICS1*-generated SA compensates fully for R493A weak function in *RRS1S RPS4* ETI.

In the *Pst AvrRpt2* growth assays, R493A in *eds1-2 sid2-1* also exhibited intermediate susceptibility between cEDS1 *eds1-2 sid2-1* and *eds1-2 sid2-1* (Fig. 6c). These data show that the same EDS1 EP-domain surface functions in bacterial resistance mediated by a TNL receptor pair and a CNL receptor, and that in both ETI systems the weak resistance signalling activity of EDS1[R493A] is buffered by *ICS1*/SA. Significantly, after stripping away SA and COR/JA effects, there was residual *EDS1*-controlled resistance in R493A *eds1-2 sid2-1* compared to *eds1-2 sid2-1* plants (Fig. 6b). Because this remaining resistance in R493A is independent of SA/COR effects in TNL ETI, we propose it is a further resistance output (sector X) regulated by the EDS1 EP domain (Fig. 7).

## Discussion

*Arabidopsis* EDS1 heterodimers, formed principally by the N-terminal lipase-like domains, are required for TNL-triggered ETI and basal immunity against host-adapted bacterial (*Pst*) and oomycete (*Hpa*) pathogens[39–41,43,51]. Current evidence suggests that the partner lipase-like domains with characteristic α/β-hydrolase folds stabilise the heterodimer, bringing together the

essential α-helical EP-domains to form a cavity[42,43]. Here, using structure-guided mutants and reductionist genetic approaches, we identify conserved positively charged residues (K440/441, K478 and R493) lining the EDS1 EP-domain cavity which control TNL immunity signalling beyond heterodimer formation. Analysis of EDS1[R493A] defects in TNL ETI against COR-producing bacteria shows that the EP-domain ensures rapid transcriptional mobilisation of host immune response pathways and timely blocking of bacterial COR-mediated repression of EDS1-induced immunity genes. We further establish that the same EDS1 EP-domain surface lining the cavity is utilised by a CNL receptor (RPS2) in ETI, and as in the TNL bacterial response, CNL resistance is made up of three genetically separable resistance sectors.

We focused on a single conserved, positively charged EP-domain residue, EDS1[R493], because mutation of this to a neutral alanine (R493A) caused complete loss of TNL immunity to *Pst AvrRps4* and two *Hpa* strains, EMWA1 and CALA2 (Fig. 1b, c, e), indicating its broad importance for TNL ETI. A positive EDS1[R493] charge rather than the arginine *per se* is required for ETI because an R493K exchange behaved as wild-type EDS1, and R493E as the null *eds1-2* mutant, in *Pst AvrRps4* and *Hpa* EMWA1 infection assays (Fig. 5a, b). Thus, altering the charge at position R493 disables EDS1 heterodimer function without disturbing its interfaces (Fig. 5a, b, Supplementary Fig. 5B). Retained partner interactions of EDS1[R493A], EDS1[R493K] and EDS1[R493E]

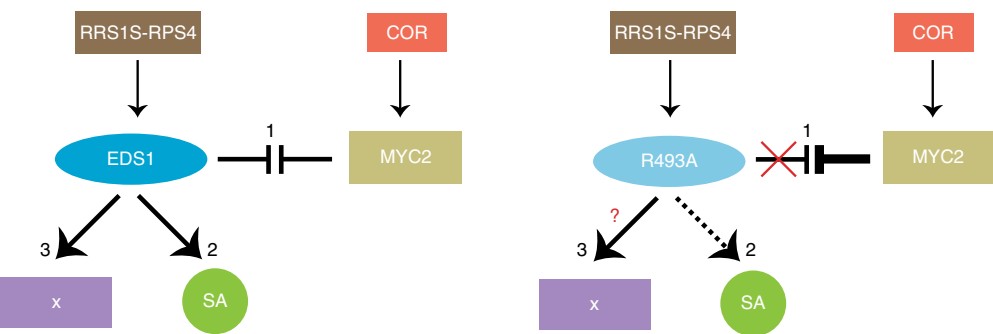

**Fig. 7** A model of EDS1 signalling branches in RRS1S RPS4 ETI. A three-pronged ETI signalling model derives from comparisons of wild-type EDS1, EDS1[R493A] and eds1-2 phenotypes in this study. Three interconnected EDS1 outputs contribute to robust TNL ETI: (1) TNL-activated wild-type EDS1 effectively counters COR antagonism of immunity gene expression via MYC2. The defective EDS1 EP-domain mutant R493A is susceptible in TNL ETI against *Pst AvrRps4* due to its inability to counter COR/MYC2 antagonism. (2) Wild-type EDS1 boosts SA accumulation independently of antagonising MYC2 while EDS1[R493A] delays SA accumulation (dashed lines) independently of COR repressive effects. (3) An additional EDS1 branch (X) in TNL (*RRS1S RPS4*) ETI is revealed after removing *ICS1*/SA and COR effects. The EDS1 EP-domain, and more specifically EDS1[R493], is also required for this resistance branch. The nature of branch X requires further study

variants (Supplementary Fig. 5A, B) and molecular modelling suggest that the charge differences do not destroy integrity of the EP-domain but rather modify ionic interactions in the cavity. Mutation of PAD4[R420] with a highly similar sequence and structural environment to EDS1[R493] but outside the cavity, did not compromise TNL ETI (Fig. 1d, e), suggesting that the location and charge of EDS1[R493] within the EP-domain cavity are crucial for EDS1-PAD4 signalling. We anticipate that functional interactions between the heterodimer EP-domains and other proteins or molecules are determined by the charge status of EDS1[R493] and other positive residues lining the cavity (Supplementary Fig. 1C). The R493A transgenic lines #1 and #2 accumulated less EDS1-YFP protein than a control cEDS1 line (Supplementary Fig. 2C, 3A, 6A, B). We established that R493A lower accumulation does not account for its defect in TNL immunity because EP-domain K487A and K387A mutant lines, as well as R493K line #1, with similar EDS1 protein levels as R493A, expressed full TNL immunity to *Pst AvrRps4* and *Hpa* EMWA1 (Fig. 5a, b and Supplementary Fig. 2D, E, 6A, B). Additionally, EDS1[R493A] expressed from a genomic construct in gR493A lines was as abundant as cEDS1 and yet fully defective in TNL ETI (Supplementary Fig. 3). We cannot exclude that low accumulation of EDS1[R493E] protein with a negative charge at residue R493 (Supplementary Fig. 6A, B) is below the threshold needed for a working EDS1 pool, although in a previous study very low amounts of nuclear-enriched EDS1-YFP protein were found to be sufficient for TNL ETI against *Pst AvrRps4* and *Hpa* EMWA1[36].

A striking feature of EDS1[R493A] is that loss of TNL immunity to *Pst AvrRps4* is conditional on bacterial COR signalling via MYC2, as indicated by *Pst AvrRps4* and *Pst Δcor AvrRps4* growth differences in R493A transgenic lines and fully restored resistance to *Pst AvrRps4* in an R493A eds1-2 myc2-3 background (Fig. 3a, b and Fig. 5b). By contrast, EDS1[R493A] behaves as a complete loss-of-function mutation in basal resistance to virulent *Pst DC3000* or *Pst Δcor* (Fig. 6a). Thus, TNL (*RRS1S RPS4*) effector recognition and/or activation[13,14,52] appears to equip EDS1 complexes, via the EP-domain positively charged surface, to block COR/MYC2-stimulated *Pst* growth (Figs. 3a, 5b).

Loss of TNL (*RPP4*) immunity to *Hpa* EMWA1 in R493A was not recovered by mutation of *MYC2* (Fig. 3c). Therefore, EDS1 heterodimers via the EP-domains likely mobilise other antimicrobial pathways or processes against *Hpa* independently of antagonising MYC2 signalling. The *Arabidopsis* mutant combinations characterised here will help to identify which pathways or

sectors are responsible for restricting *Hpa* growth in TNL ETI, building on earlier gene expression microarray analyses of ETI responses to *Hpa* isolates[53,54]. A number of protein effectors delivered to plant host cells by *Hpa*, fungi and *P. syringae* bacteria suppress SA immunity by targeting JA signalling[24,25,55–57]. For example, *P. syringae* effectors HopX1 and HopZ1 derepress JA response genes independently of bacterial COR[55,57]. *P. syringae* effector HopBB1 activates a subset of JA outputs by increasing the binding between two JA response repressors, JAZ3 and a TF TCP14, leading to their COI1-mediated elimination[25]. *Hpa* effector HaRxL44 interacts with and degrades Mediator subunit 19 A to suppress SA signalling[56]. Since, EDS1 complexes effectively block bacterial COR virulence early in *RRS1S RPS4* ETI (Figs. 3a, 5b) and in our RNA-seq analysis there is a significant overlap between TNL/*EDS1*-dependent DEG at 8 hpi and SA/JA-responsive genes (Supplementary Fig. 4B), we speculate that an important EDS1 heterodimer function in ETI is to preserve SA-based immunity against interference with the phytohormone network by effectors from multiple pathogens.

A further insight to TNL/EDS1 ETI against *Pst AvrRps4* bacteria gained from this analysis is that perturbation of the EDS1 EP-domain in R493A lines causes a general delay in transcriptional reprogramming of defence pathways, which, although not caused by bacterial COR, renders the plant vulnerable to COR disease-promoting effects. This is seen most clearly at the level of free (active) SA accumulation at 8 hpi and 24 hpi (Figs. 2a, 4a) and in RNA-seq analyses of wild-type Col (or cEDS1), eds1-2 and R493A responses at 8 hpi and 24 hpi with *Pst AvrRps4* and *Pst Δcor AvrRps4* (Figs. 2c, 4b, c). Partial recovery of gene expression changes in R493A lines at 24 hpi with *Pst AvrRps4* without restoration of *RRS1S RPS4* immunity (Figs. 2, 4b) further emphasizes that there is a critical expression time-window for ETI to succeed. Hence, early TNL/EDS1 induction of the *ICS1*/SA resistance branch and probably numerous other anti-microbial outputs (before or at 8 hpi) is important for TNL ETI against COR-producing *Pst AvrRps4* (Figs. 3a, 6b). This might also be the case for TNL-recognised *Hpa* strains (Fig. 1b, e), although the time-frame for transcriptional activation of ETI responses to *Hpa* strains appears to be more protracted[53,54]. It remains to be established whether timely defence gene expression reprogramming mediated by the EDS1 EP-domain is also necessary for TNL immunity against *Hpa* strains.

Based on the above data, we propose that effective TNL/EDS1 immunity involves a critical early step for rapid transcriptional

activation of a broad set of defence pathways, rather than selectively mobilising resistance outputs. This makes sense because ETI and basal immune responses differ principally in timing and amplitude of transcriptional reprogramming, while the topology of co-expression networks appears to be quite stable[5,8,9,49,58]. Moreover, ETI specifically triggered by one pathogen effector or strain confers broad-spectrum immunity to a range of pathogen types[59].

By comparing *Pst AvrRps4* and *Pst Δcor AvrRps4* transcriptomes in a weakly active EDS1[R493A] mutant we have uncovered an interesting set of immune-related DEG (cluster #17) (Fig. 4c). These genes would not have emerged from analysis of cEDS1 and null *eds1-2* mutant responses alone because they are repressed at 8 hpi only with *Pst AvrRps4* in R493A (Fig. 4c). Cluster #17 contains a number of functionally defined *NLR* receptor and *WRKY* TF genes, and one member (*ADR1-L2*) of a family of conserved helper NLRs (Fig. 4c), which contribute to ETI[60–62]. Most of the cluster #17 genes are not represented in 'ETI-related' gene sets extracted from *RPS2* (CNL) transcriptomic analyses (Supplementary Fig. 4C)[9,49]. Therefore, cluster #17 might represent a new level of plant host gene expression control in TNL ETI in which key immunity components are protected from pathogen repression. It is tempting to speculate that susceptibility of R493A to *Hpa* (Fig. 1b, c) is due in part to a failure to block *Hpa* effectors from targeting some or all of these genes for repression.

Analysis of EDS1[R493A] reinforces a two-pronged EDS1 transcriptional mechanism in *RRS1S RPS4* ETI–(i) promoting *ICS1* generated SA and (ii) blocking COR/MYC2 suppression of SA immunity[19] (Fig. 7). Delayed free SA accumulation in R493A plants at 8 hpi with *Pst AvrRps4* and *Pst Δcor AvrRps4* (Fig. 4a) highlights a failure of this weak EDS1 allele to rapidly mobilise the SA-branch independently of COR. Moreover, early antagonism of COR/MYC2 signalling is important in *RRS1S RPS4* ETI because EDS1[R493A] is fully susceptible to *Pst AvrRps4* despite being able to suppress COR/MYC2 marker genes almost as well as cEDS1 later at 24 hpi (Figs. 2c, 4c). It is significant that EDS1[R493A] retains a portion of TNL resistance to *Pst Δcor AvrRps4* after removal of the *ICS1*/SA branch in an *eds1-2 sid2-1* double mutant (Fig. 6b). We therefore add a third EDS1 TNL immunity sector (denoted X in Fig. 7) which is independent of SA and JA/COR crosstalk. By testing R493A *eds1-2 sid2-1* plants in ETI conferred by CNL receptor *RPS2* recognising *Pst AvrRpt2*[44,50] (Fig. 6c), it becomes clear that the EDS1 EP-domain cavity surface signals in bacterial ETI triggered not only by TNL receptors but at least one CNL receptor type. This has implications for understanding plant NLR receptor molecular functions because the N-terminal TIR and CC domains are structurally different and therefore unlikely to be solely responsible for signalling via the EDS1 EP-domain in bacterial ETI[4].

## Methods

**Plant materials, growth conditions and pathogen strains.** All mutants and lines are in *Arabidopsis* accession Col-0. The mutants *eds1-2*, *sid2-1*, *eds1-2 sid2-1*, *myc2-3*, *pad4-1 sag101-3*, *eds1-2 myc2-3*, as well as YFP-cEDS1 and gEDS1-YFP transgenic lines were previously described[19,35,43]. *Pseudomonas syringae* pv. *tomato* (*Pst*) strain DC3000, *Pst Δcor*, *Pst* DC3000 *AvrRps4* (*Pst AvrRps4*), DC3000 *Δcor AvrRps4* (*Pst Δcor AvrRps4*) and DC3000 *AvrRpt2* (*Pst AvrRpt2*) are described[17]. Plants were grown on soil in controlled environment chambers under a 10 h light regime (150-200 µE/m²s) at 22 °C and 60% relative humidity.

**Pathogen infection assays.** For bacterial growth assays, *Pst AvrRps4* or *Pst Δcor AvrRps4* (OD$_{600}$ = 0.0005) in 10 mM MgCl$_2$ were hand-infiltrated into leaves of 4-week-old plants and bacterial titres measured at 4 h post infiltration (day 0) and day 3 as described[41]. Each biological replicate consists of three leaf disks from different plants and data shown in each experiment is compiled from 3–4 biological replicates. Statistical analysis was performed using one-way ANOVA with multiple testing correction using Tukey's HSD ($p < 0.005$).

For gene expression and protein accumulation assays, leaves from 4-week-old plants were hand-infiltrated with bacteria (OD$_{600}$ = 0.005) and samples taken at indicated time points. For measuring protein accumulation, samples were pooled from at least three different plants. For gene expression analysis by qRT-PCR, four or more leaves from different plants were pooled as one biological replicate and two biological replicates were used in each independent experiment. Statistical analysis was performed by Student's *t*-test with multiple testing correction using Bonferroni method ($p < 0.05$).

*Hpa* isolates EMWA1 and CALA2 were sprayed onto 2–3 week-old plants at $4 \times 10^4$ spores/ml dH$_2$O. Plant host cell death and *Hpa* infection structures were visualised under a light microscope after staining leaves with lactophenol trypan blue as described[63]. T$_1$ complementation assays of *Arabidopsis* transgenic lines were performed as previously described[45] and *Hpa*-infected seedlings rescued by spraying with Ridomil Gold (Syngenta). To quantify *Hpa* sporulation on leaves, three pots of each genotype were infected and treated as biological replicates. Plants were harvested at 6 dpi, their fresh weight determined, and conidiospores suspended in 5–10 ml dH$_2$O and counted under the microscope using a Neubauer counting chamber.

**RNA isolation, library preparation and sequencing.** For RNA-seq experiments, Col, *eds1-2*, cEDS1 and R493A#1 4-week-old plants were infiltrated with *Pst AvrRps4* or *Pst Δcor AvrRps4* using the same bacterial titre as for gene expression assays. To randomise samples and reduce variation, total RNA was isolated from four individual plants per genotype (three infected leaves per plant) and pooled as one biological replicate. Each biological replicate was from an independent experiment. Total RNA was purified with an RNeasy Plant Mini Kit (Qiagen) according to manufacturer's instructions. RNA-seq libraries were prepared from 1 µg total RNA according to TruSeq RNA sample preparation v2 guide (Illumina). Library construction and RNA sequencing was done by the Max-Planck Genome Centre (MPIPZ, Cologne), and produced 21–32 million 100 bp single-end reads per sample. RNA-seq reads were mapped to the annotated genome of *Arabidopsis thaliana* (TAIR10) using TOPHAT2 ($a = 10$, $g = 10$)[64] and transformed into a read count per gene per sample using the htseq-count script (s = reverse, t = exon) in the package HTSeq[65]. Genes with <100 reads across samples were discarded. Count data from the remaining genes were TMM-normalised and log$_2$ transformed using functions 'calcNormFactors' (R package EdgeR[66]) and 'voom' (R package limma[67]). To analyse differential gene expression over time between the different genotypes and treatments, for each analysis we fitted a linear model to the respective log$_2$-transformed count data using the function lmFit (R package limma[67]) and subsequently performed moderated *t*-tests for specific comparisons of interest. In all cases, the resulting *p*-values were adjusted for false discoveries due to multiple hypothesis testing via the Benjamini-Hochberg procedure. For each comparison, we extracted a set of significantly differentially expressed genes between the tested conditions (adjusted *p*-value ≤ 0.05, |log2FC| ≥ 1). RNA-seq experiments for *Pst AvrRps4* and *Pst Δcor AvrRps4* were performed in separate batches and therefore normalised to Col for the respective treatments to negate potential batch effects. The normalised values were used to generate a heatmap with hierarchial clustering. Circos plot was created using the R package 'Circlize'[68], to show the overlap of cluster #17 with other datasets.

**qRT-PCR analysis.** Total RNA was extracted using a Plant RNA kit (Bio-budget). Five hundred nanogram total RNA was used for cDNA synthesis (quanta bio) and qRT-PCR analysis was performed using SYBR green master mix. The housekeeping gene *GapDH* was used as reference.

**Plasmid constructs.** The pENTR/D-TOPO-cEDS1, pENTR/D-TOPO-gEDS1 and pENTR/D-TOPO cPAD4 vectors used for site-directed mutagenesis are previously described[36,43]. Site-directed mutagenesis on the entry vectors was performed according to the QuikChange II site-directed mutagenesis manual (Agilent). Mutated entry clones were verified by sequencing and recombined into a pAM-PAT-based binary vector backbone by LR reaction.

**Generation of *Arabidopsis* transgenic plants.** Stable transgenic lines were generated by transforming binary expression vectors into *Arabidopsis* null mutants *eds1-2* or *pad4-1 sag101-3*, as indicated, using *Agrobacterium*-mediated floral dipping.

**Yeast two-hybrid assays.** Yeast 2-hybrid (Y2H) assays were performed using the Matchmaker system (Clontech) with strain AH109. Gateway cassettes were cloned into pGADT7 and pGBKT7 plasmids. Newly made PAD4 and EDS1 site-directed EP-domain mutants and the EDS1[LLIF] variant[43] were recombined into these plasmids by LR reaction. pGAD-containing and pGBK-containing co-transformants were selected on plates lacking leucine and tryptophan (-LW). Single colonies were re-streaked on plates additionally lacking histidine and adenine (-LWHA) to monitor reporter activation. Yeast growth was recorded after 2–5 days incubation at 30 °C.

**Transient expression in *Arabidopsis* protoplasts**. Leaf mesophyll protoplasts were prepared from 4-week-old *eds1-2 pad4-1 sag101-3* plants and transfected with plasmid DNA[69]. After transfection, protoplasts were incubated at room temperature under weak light (1.5 μE/m²s) for 16 h. Protoplasts were harvested and IPs performed as described below.

**Protein extraction, immunoprecipitation, immunoblotting**. Total leaf extracts or protoplasts were processed in extraction buffer (50 mM Tris pH 7.5, 150 mM NaCl, 10% (v/v) glycerol, 2 mM EDTA, 5 mM DTT, protease inhibitor (Roche, 1 tablet per 50 ml, 0.1% Triton). Lysates were centrifuged for 15 min, 12,000 rpm at 4 °C. 50 μl of supernatant was used as input sample. Immunoprecipitations (IPs) were conducted by incubating the input sample with 12 μl GFP-TrapA beads (Chromotek) for 2 h at 4 °C. Beads were collected by centrifugation at 2000 rpm, 1 min at 4 °C. Beads were washed three times in extraction buffer and boiled at 95 °C in 2× Laemmli buffer for 10 min. Proteins were separated by SDS-PAGE and analysed by immunoblotting. Antibodies used were α-GFP (1:5000; Sigma Aldrich, 11814460001), α-HA (1:5000; Sigma Aldrich, 11867423001), α-FLAG (1:5000; Sigma Aldrich, F3165). Secondary antibodies coupled to Horseradish Peroxidase (HRP) were used for protein detection on blots (1:5000; Sigma Aldrich, 005000000005295191).

**SA quantitation**. Free SA was quantified from leaf tissues (70–200 mg fresh weight), of 4-week-old plants using a chloroform/methanol extraction and analysed by gas chromatography coupled to a mass spectrometer (GC-MS, Agilent)[70]. Statistical analysis was performed by Student's *t*-test with multiple testing correction using the Bonferroni method ($p < 0.05$).

**Statistical analysis**. Three independent experiments were performed for all assays unless otherwise indicated. The number of biological replicates used for each experiment shown isindicated in the corresponding figure legend. Statistical significance was determined either using a Student's *t* test or one-way ANOVA. Bonferroni or Tukey's HSD Posthoc tests for multiple variate analysis were applied as appropriate.

**Reporting summary**. Further information on experimental design is available in the Nature Research Reporting Summary linked to this article.

## Data availability

Source data are provided as a Source Data file. The RNA-seq data are deposited in the National Center for Biotechnology Information Gene Expression Omnibus (GEO) database with accession number GSE116269.

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

## Acknowledgements

We thank Kenichi Tsuda and Takaki Maekawa (MPIPZ Cologne) for helpful discussions. This work was supported by The Max-Planck Society and Deutsche Forschungsgemeinschaft (DFG) grants within SFB 635 and SFB 670 (JEP, DDB), SFB 680 (JEP, DL), and an International Max-Planck Research School (IMPRS) doctoral fellowship (PvB).

## Author contributions

D.D.B. and J.E.P. designed the study; D.D.B. and P.V.B. performed experiments; B.K., D. D.B. and D.L. analysed the RNA-seq data; D.D.B. and J.B. generated transgenic plant lines; K.N. provided structural insights; D.D.B. and J.E.P. wrote the paper with inputs from D.L.

## Additional information

**Competing interests:** The authors declare no competing interests.

