## [Peer Review File · Nature Communications]

Reviewers' comments:

Reviewer #1 (Remarks to the Author):

Plants rely on an innate immune system to detect and defend against all potential pathogens. One of the major components of this system are nucleotide-binding, leucine-rich repeat (NLR) immune receptors which detect evolved pathogens by the presence or activity in the host cytoplasm. Despite their importance, how these immune receptors function to signal downstream remains largely unknown. The EDS1 protein is required for the function of all tested TIR-NLR proteins, and is a candidate signaling protein. EDS1 is critical for many immune responses, but is also poorly understood. EDS1 is known to heterodimerize with related proteins (such as PAD4 and SAG101), and a EDS1-SAG101 cocrystal suggests that N-terminal dimerization via the (catalytically-inactive) lipase-like domain promotes dimerization of the C-terminal EP domain. The function of the EP domain is unknown, but in this paper the authors look at a pocket formed in the hetero-EP dimer that contains several charged residues (R493 being closely examined). They find R493 is required for EDS1 function (but does not disrupt EDS1-PAD4 heteroassociation). This represents a candidate signaling structure, and thus an important next step in understanding TIR-NLR signaling.

The paper is largely well-done and represents an important step forward for understanding EDS1 function. However, the paper suffers from a fatal flaw that the authors are aware of, but don't fully address.

The authors generate two transgenic lines where *eds1/pad4* is complemented with a cDNA clone of EDS1 or EDS1 R493A. In Figure S2D, a western blot for cDNA line #1 shows that EDS1 R493 doesn't accumulate well. Protein accumulation for line #1 is not quantified, but it is dramatically lower (protein accumulation of cDNA line #2 is not shown). The authors rightly state that it is not possible to interpret loss of function phenotypes for a transgenic that doesn't accumulate protein. They make a genomic EDS1 R493A line, show that it accumulates more similar levels to WT genomic EDS1 (and dramatically more than the WT cDNA). They test this genomic line and (like the cDNA R493A line) it is a loss of function for *avrRPS4* ETI. Thus, it is reasonable to conclude that R493A in the genomic context is a loss of function for *avrRPS4* ETI due to something specific to the mutation and not due to a mere lack of protein accumulation.

The fatal flaw is that the cDNA R493A lines are still uninterpretable due to lack of accumulation, and the authors exclusively use the cDNA lines in all the other figures of the paper. The fact that R493A in the genomic context gives an interpretable result doesn't make the weakly accumulating cDNA lines any more usable.

Reviewer #2 (Remarks to the Author):

The manuscript by Bhandari et al. provides important new information on the central plant immune regulator ENHANCED DISEASE SUSCEPTIBILITY1 (EDS1). EDS1 interacts with two other proteins of similar domain structure, PAD4 and SAG101, to regulate multiple plant defense responses. With the recently resolved crystal structure of the EDS1-SAG101 and the modeled EDS1-PAD4 heterodimers, the authors focus on a cavity formed by the EDS1 C-terminal EP domain that is lined with positive charges. A series of elegant experiments demonstrate that mutations in one positively charged amino acid in particular, R493, effectively separates different functions of EDS1. Consequently, the authors propose that EDS1 regulates three major downstream networks in effector-triggered immunity: upregulation of the SA sector, counteracting activation of the JA-response pathway by the bacterial toxin coronatine, and a third uncharacterized sector that is not limited to the class of immune receptors usually associated with EDS1 function.

The authors present a wealth of data, and the level of detail at which Parker and her group can dissect and analyze EDS1 function is impressive. One important piece that is missing so far is the molecular function of the EP domain. Presumably this domain is a protein-protein interaction domain, with positive charges determining proper interactions, but this is not addressed experimentally. Nevertheless, this manuscript provides novel insights, new tools and profound insights into the molecular nature of plant immune response sectors. While these sectors have been implied and modeled, the work here establishes a novel opening that should lead to further characterization of plant immune mechanisms.

Minor points:

- 1) line 40: replace "basal immunity" with "MTI/PTI". These are not equivalent.
- 2) 64 and throughout manuscript: wild-type Arabidopsis protein names are written in all-caps
- 3) Figure 3: in experiments with Hpa EMWA1 (C), include the eds1-2 cEDS1 and eds1-2 myc2-3 cEDS1 lines to demonstrate that cEDS1 fully complements the eds1-2 myc2-3 line.

Reviewer #3 (Remarks to the Author):

In the manuscript "an EDS1 heterodimer signalling surface enforces timely reprogramming of immunity genes in Arabidopsis" Bhandari and co-authors make an important contribution to the understanding of NLR immune signaling by carefully dissecting the roles of a positively charged cavity in the EDS1 interface with its interaction partners PAD4 or SAG101. Here the EDS1 EP-domain and its positively charged cavity are demonstrated to play a role in at least three distinct resistance pathways including the ETI specific responses shared by CNL and TNL NLRs and the counteracting of bacterial Coronatin-mediated suppression of immunity. In my opinion the data and well thought-out experiments support strongly the claims and the paper is well written. There is little known on the signaling steps immediately downstream from NLRs and how NLR activation is linked to transcriptional reprogramming. This manuscript presents an important step in the thinking in the field of plant resistance signaling.

I do have some suggestions for improving the manuscript.

Minor comments

Line 71: 'EP' domains/ EP-domains > abbreviation for EDS1-PAD4?

Line 77: antagonism of the COR/JA MYC2-branch > the interaction of PAD4 and MYC2 is not mentioned. Some information on that interaction would make it easier for the reader to understand Fig S4 B and C.

Line 94 & Fig S1: several conserved positively charged amino acids > A few comments 1. I feel it would be better to give the exact number of conserved positively charged amino acids in the text. And is the exact identity conserved or the charge? (this might be relevant for the interpretation of the results with the R493K mutant) 2. The cavity is formed by surfaces on both partners in the heterodimer. It is not clear from the text or the figures what the contribution of each partner is, only a subset of the residues in EDS1 are studied by mutagenesis; are the positively charged residues on the PAD4/SAG101-side of the cavity also conserved? 3. Fig S1 shows the cavity between EDS1 and SAG101 with the positively charged residues shown as sticks, but without labels, so it is not easy to relate the residues mentioned in S1B, C and D to the structure in S1A. 4. It would be interesting to see an image with the net surface charge in the cavity. Does the cavity have an overall positive charge or are there negatively charged residues present that partially compensate for the positive charge? 5. In Fig 1D conserved residues in the EP-domains are represented as spheres; I think it is good to list these residues.

Line 154 & Fig S2B: to conclude that the nucleocytoplasmic fluorescence was lower for the mutant requires controlled confocal settings (imaging under identical laser power, gain, etc.). I assume

this was the case, but I think that it should be mentioned somewhere (main text or figure legend).

Line 199: ... with 5993 DEG between eds1-2 and cEDS1 (Fig. 2C). > Figure 2C clearly shows that EDS1 R493A gives a delayed transcriptional reprogramming. The overlap or lack of overlap between the number of DEGs in the comparisons of R493A vs eds1-2, R493A vs cEDS1 and eds1-2 vs cEDS1 is mentioned in the text, but maybe a small Venn diagram would help to make this clearer.

line 204; Hence, the EDS1 EP-domain lining the heterodimer cavity is critical > Does the author mean the EDS1 EP-domain residues lining the cavity? I think only conclusions can be drawn about R493 specifically and not the other residues lining the cavity.

Line 274/Fig 4C: the labeling of this figure is small and difficult to read. Especially the sector numbers (#17 is printed a bit larger and is readable). The Pst lines Pst AvrRps4 and Pst DeltaCor AvrRps4 are indicated by Avr and DeltaCor in the labeling in this and other figures. If you just quickly scan the image it might seem that the DeltaCor lacks the Avr, whereas both are identical except for the lack of Cor. Maybe Avr Cor / Avr DeltaCor would be clearer.

Line 286 & 324: BTH > abbreviation in full. Maybe good to mention it is an SA-analogue in the legend of Fig 4 too.

Line 347 & Fig S4 C: A few comments. 1. there appears to be a small mistake in the labeling of the proteins in the second lane from the left; now it shows that both EDS1-FLAG and EDS1- LIFF-FLAG are expressed in the same lane. 2. The text in line 347-348 is a bit brief on the nature of the interactions between the EDS1, PAD4 and MYC2 proteins; the earlier finding that the interaction of EDS1 to PAD4 suppresses the interaction of PAD4 with MYC2 (Cui et al., (2018) Molecular Plant 11(8): 1053-1066) is not mentioned. That explains why in Fig S4 C the highest level of MYC2 is co-IPed with PAD4 in the presence of the EDS1 LLIF mutant. The coexpression of EDS1 leads to a lower MYC2 interaction and indeed a similar effect is seen for the three R493 mutants.

line 538; appears to be dictated by something else than structurally different TIR and CC receptor signaling domains. > may need a bit clearer wording. The assumption is made that because the CC and TIR domains have different structures they are unlikely to be the NLR domains responsible for signaling this sector X response?

Line 566: performed either using... > there appears to be something missing in this sentence.

Response to Editor and reviewers: NCOMMS-18-22434, Bhandari et al.

Response (in BOLD) to Reviewer specific comments:

Reviewer #1

The authors generate two transgenics lines where *eds1/pad4* is complemented with a cDNA clone of EDS1 or EDS1 R493A. In Figure S2D, a western blot for cDNA line #1 shows that EDS1 R493 doesn't accumulate well. Protein accumulation for line #1 is not quantified, but it is dramatically lower (protein accumulation of cDNA line #2 is not shown). The authors rightly state that it is not possible to interpret loss of function phenotypes for a transgenic that doesn't accumulate protein. They make a genomic EDS1 R493A line, show that it accumulates more similar levels to WT genomic EDS1 (and dramatically more than the WT cDNA). They test this genomic line and (like the cDNA R493A line) it is a loss of function for *avrRPS4* ETI. Thus, it is reasonable to conclude that R493A in the genomic context is a loss of function for *avrRPS4* ETI due to something specific to the mutation and not due to a mere lack of protein accumulation. The fatal flaw is that the cDNA R493A lines are still uninterpretable due to lack of accumulation, and the authors exclusively use the cDNA lines in all the other figures of the paper. The fact that R493A in the genomic context gives an interpretable result doesn't make the weakly accumulating cDNA lines any more usable.

Response: Thanks for the positive comments and critical feedback. Yes we recognized the possibility that resistance and transcriptional defects in the cR493A lines might be explained by lower protein accumulation. Our data argue against this because i) the gR493A line with higher protein has equivalent susceptibility to *Pst AvrRps4* as cR493A (original Fig. S2 C & D), ii) that cR493A #1 and #2 susceptibility to *Pst AvrRps4* is conditional on COR and MYC2 (original Figs 3A, B and 4). In this revision we present additional experiments which further support our claim that the cR493A signalling defect is not due to its low protein accumulation. In revised Fig. S2 we compare resistance to *Pst AvrRps4* between cEDS1 and R493A lines #1 and #2 with two independent homozygous transgenic lines (#1 and #2) of the cDNA mutant K487A (within the EP-cavity like R493A) and K387A (outside the EP-cavity) which were resistant to *Hpa* (original Fig. S1C). These lines are also fully resistant to *Pst AvrRps4* (new Fig. S2C). In a Western blot analysis of the cEDS1 mutant lines, K487A and K387A protein accumulation is similar or less than that of R493A (new Fig. S2D).

Based on the original cR493A vs gR493A data and the fact that cR493A lines were conditionally susceptible to *Pst AvrRps4* (+COR), we selected the cR493A lines for detailed phenotyping and cR493A line #1 for RNA-seq. In a further set of experiments, we have tested whether gR493A lines #1 and #2 with higher EDS1 protein accumulation respond differently to cDNA R493A lines #1 and #2 at the level of *Pst AvrRps4*-induced gene expression at 8 hpi (- the key time point for differential gene expression changes between R493A and cEDS1 or Col). Data in new Fig. S3 A-C show that cR493A and gR493A lines

have equivalent *Pst AvrRps4* resistance and gene expression profiles for SA-dependent genes *ICS1* and *PBS3*, an SA-independent gene (*FMO1*) and an EDS1-repressed gene (*MYB34*). Therefore, gR493A higher protein accumulation does not recover the capacity for timely gene expression reprogramming, which we would expect it to do if the cR493A defect is impacted by lower EDS1 protein levels. Considering all of these data together, we think our point that the observed R493A signalling defects are not because of low protein accumulation is strongly supported.

Reviewer #2

The authors present a wealth of data, and the level of detail at which Parker and her group can dissect and analyze EDS1 function is impressive. One important piece that is missing so far is the molecular function of the EP domain. Presumably this domain is a protein-protein interaction domain, with positive charges determining proper interactions, but this is not addressed experimentally. Nevertheless, this manuscript provides novel insights, new tools and profound insights into the molecular nature of plant immune response sectors. While these sectors have been implied and modeled, the work here establishes a novel opening that should lead to further characterization of plant immune mechanisms.

Response: Thank you for the positive comments and critical feedback. We have tested various protein candidates for direct interaction mediated by the EDS1-PAD4 heterodimer EP domain but so far have not found a factor which fits the bill of binding physically via this interface. We are still actively researching this. We appreciate that the reviewer nevertheless thinks that the data presented here provide major new insights to the nature of the plant immunity response.

Minor points:

1) line 40: replace "basal immunity" with "MTI/PTI". These are not equivalent.

Thanks, this is corrected.

2) line 64 and throughout manuscript: wild-type Arabidopsis protein names are written in all-caps

Nat Comms published articles we've viewed use lower case for protein full names so we have stayed with that. We can readily change this if required. We've italicized *Arabidopsis*.

3) Figure 3: in experiments with Hpa EMWA1 (C), include the eds1-2 cEDS1 and eds1-2 myc2-3 cEDS1 lines to demonstrate that cEDS1 fully complements the eds1-2 myc2-3 line.

Thanks for pointing out. We've added *Hpa* spore count data in the eds1-2 cEDS1 and eds1-2 myc2-3 cEDS1 lines to Fig. 3C. As with the *Pst AvrRps4* data, our cEDS1 line fully complements in resistance to *Hpa Emwa1*.

Reviewer #3

In the manuscript "an EDS1 heterodimer signalling surface enforces timely reprogramming of immunity genes in Arabidopsis" Bhandari and co-authors make an important contribution to the understanding of NLR immune signaling by carefully dissecting the roles of a positively charged cavity in the EDS1 interface with its interaction partners PAD4 or SAG101. Here the EDS1 EP-

domain and its positively charged cavity are demonstrated to play a role in at least three distinct resistance pathways including the ETI specific responses shared by CNL and TNL NLRs and the counteracting of bacterial Coronatin-mediated suppression of immunity. In my opinion the data and well thought-out experiments support strongly the claims and the paper is well written. There is little known on the signaling steps immediately downstream from NLRs and how NLR activation is linked to transcriptional reprogramming. This manuscript presents an important step in the thinking in the field of plant resistance signaling. I do have some suggestions for improving the manuscript.

Thanks for all comments below. They've been very helpful.

Minor comments

Line 71: 'EP' domains/ EP-domains > abbreviation for EDS1-PAD4?

Yes. We've changed text. We also deposited the EDS1 EP-domain as a unique domain in PFAM (PF18117), https://pfam.xfam.org/family/EDS1_EP (ln 71)

Line 77: antagonism of the COR/JA MYC2-branch > the interaction of PAD4 and MYC2 is not mentioned. Some information on that interaction would make it easier for the reader to understand Fig S4 B and C.

A good point. We write in Introduction (ln 77): 'PAD4 was found to interact with MYC2 *in planta* likely by indirect association. EDS1-PAD4 antagonism of the COR/JA MYC2-branch was nuclear, coincident with or after MYC2 release from COI1-JAZ nuclear complexes, and independent of EDS1-PAD4 promotion of the ICS1/SA-branch'.

We expand on the MYC2 antagonism in Results (in response to Rev. #3 point 11).

1. Line 94 & Fig S1: several conserved positively charged amino acids > A few comments 1. I feel it would be better to give the exact number of conserved positively charged amino acids in the text. And is the exact identity conserved or the charge? (this might be relevant for the interpretation of the results with the R493K mutant)

We now highlight 10 EDS1 positively charged residues lining the cavity in Fig. S1B. Of these, Fig. S1B shows that four arginines (R) and a histidine (H) are strictly conserved across seed plants. One arginine (R) and three lysines (K) have a conserved charge. K487 is variable. We state in text (ln 96): 'The Arabidopsis EDS1-SAG101 heterodimer crystal structure reveals a cavity formed by the partner EP-domains with nine EDS1 positively charged amino acids (three lysines, five arginines and a histidine) that are conserved across seed plants (Fig. S1A, B)⁴³.

2.The cavity is formed by surfaces on both partners in the heterodimer. It is not clear from the text or the figures what the contribution of each partner is, only a subset of the residues in EDS1 are studied by mutagenesis; are the positively charged residues on the PAD4/SAG101-side of the cavity also conserved?

The contribution of the EDS1 partners to the cavity was shown in the Wagner et al (2013) CHOM paper but specific residues were not interrogated. It was apparent from the protein sequence phylogenies we examined in Wagner et al that PAD4 and SAG101 have

some conserved positive residues in the EP domains. We are working with more recently available plant genome sequences for a more comprehensive sequence analysis of PAD4 and SAG101 orthologues and targeting residues bordering the cavity for mutational analysis. So, this is in progress but behind the EDS1 analysis.

3. Fig S1 shows the cavity between EDS1 and SAG101 with the positively charged residues shown as sticks, but without labels, so it is not easy to relate the residues mentioned in S1B, C and D to the structure in S1A.

We have labeled the sticks in Fig. S1A so clearer for linking with other Fig. S1 panels.

4. It would be interesting to see an image with the net surface charge in the cavity. Does the cavity have an overall positive charge or are there negatively charged residues present that partially compensate for the positive charge?

We have added an electrostatic (red-blue charge) representation of EDS1 to Fig. S1A which shows the overall positive nature of the EDS1 cavity surface.

5. In Fig 1D conserved residues in the EP-domains are represented as spheres; I think it is good to list these residues.

We display the precise sphere-represented residues in the respective zoom outs in Fig. 1D and explain in Fig. legend.

6. Line 154 & Fig S2B: to conclude that the nucleocytoplasmic fluorescence was lower for the mutant requires controlled confocal settings (imaging under identical laser power, gain, etc.). I assume this was the case, but I think that it should be mentioned somewhere (main text or figure legend).

The images shown were taken at different settings because at the same setting wt YFP-EDS1 became over-exposed. This suggested there was more wt YFP-EDS1 protein than YFP-R493A in the transgenic lines, which was borne out by the Western blot analysis (please also see new Fig. S2C). We rephrase in text (ln 156) and clarify this point in Fig. S2B legend.

7. Line 199: ... with 5993 DEG between eds1-2 and cEDS1 (Fig. 2C). > Figure 2C clearly shows that EDS1 R493A gives a delayed transcriptional reprogramming. The overlap or lack of overlap between the number of DEGs in the comparisons of R493A vs eds1-2, R493A vs cEDS1 and eds1-2 vs cEDS1 is mentioned in the text, but maybe a small Venn diagram would help to make this clearer.

Thanks for pointing out. In new Fig. S4A we have corresponding Venn diagrams to show overlap of gene expression changes at 8 and 24 hpi between cEDS1, R493A and eds1-2.

8. line 204; Hence, the EDS1 EP-domain lining the heterodimer cavity is critical > Does the author mean the EDS1 EP-domain residues lining the cavity? I think only conclusions can be drawn about R493 specifically and not the other residues lining the cavity.

We based this statement on the susceptibility phenotypes of several EDS1 EP-domain mutations (shown in Fig. S1C) and the more detailed characterization of R493A. But we take the reviewer's point and have rephrased (ln 201):

'Hence, R493 in the EDS1 EP-domain lining the heterodimer cavity is critical for timely TNL transcriptional defence reprogramming.'

9. Line 274/ Fig 4C: the labeling of this figure is small and difficult to read. Especially the sector numbers (#17 is printed a bit larger and is readable).

The Pst lines Pst AvrRps4 and Pst DeltaCor AvrRps4 are indicated by Avr and DeltaCor in the labeling in this and other figures. If you just quickly scan the image it might seem that the DeltaCor lacks the Avr, whereas both are identical except for the lack of Cor. Maybe Avr Cor / Avr DeltaCor would be clearer.

Agreed. We've improved the labels for all reviewer's points in Fig. 4.

10. Line 286 & 324: BTH > abbreviation in full. Maybe good to mention it is an SA-analogue in the legend of Fig 4 too.

Done.

11. Line 347 & Fig S4 C: A few comments. 1. there appears to be a small mistake in the labeling of the proteins in the second lane from the left; now it shows that both EDS1-FLAG and EDS1-LIFF-FLAG are expressed in the same lane.

1. Thanks for spotting this labelling mistake in now Fig. S5C. This is corrected now to indicate that PAD4-YFP is able to IP EDS1-FLAG but not EDS1 LLIF-FLAG.

2. The text in line 347-348 is a bit brief on the nature of the interactions between the EDS1, PAD4 and MYC2 proteins; the earlier finding that the interaction of EDS1 to PAD4 suppresses the interaction of PAD4 with MYC2 (Cui et al., (2018) Molecular Plant 11(8): 1053-1066) is not mentioned. That explains why in Fig S4 C the highest level of MYC2 is co-IPed with PAD4 in the presence of the EDS1 LLIF mutant. The coexpression of EDS1 leads to a lower MYC2 interaction and indeed a similar effect is seen for the three R493 mutants.

2. Sorry we didn't explain this well. Text is now: 'Additionally, EDS1^{R493} variants did not alter interactions between EDS1/PAD4-YFP complexes and StrepII-HA (SH)-tagged MYC2 in IPs of transiently expressed proteins (Fig. S5C), showing that R493 is not responsible for PAD4 association with MYC2. In these assays, EDS1 but not EDS1^{LLIF} (which does not bind PAD4) reduced PAD4-MYC2 IP signals (Fig. S5C). These data reinforce earlier evidence that interaction between PAD4 and MYC2 is competed by EDS1 in IPs and alone does not determine antagonism of MYC2 by the EDS1-PAD4 complex in TNL (*RRS1S RPS4*) ETI¹⁹.' (ln 355-362)

line 538; appears to be dictated by something else than structurally different TIR and CC receptor signaling domains. > may need a bit clearer wording. The assumption is made that because the CC and TIR domains have different structures they are unlikely to be the NLR domains responsible for signaling this sector X response?

A fair point. We have rephrased: ‘ This has implications for understanding plant NLR receptor functions because the N-terminal TIR and CC domains are structurally different and therefore unlikely to be solely responsible for signalling via the EDS1 EP-domain in bacterial ETI ⁴.’
We can’t assume this use of the EP domain in CNL and TNL ETI applies only to sector X, which is an interesting output to follow up on.

Line 566: performed either using... > there appears to be something missing in this sentence.

Corrected

Reviewers' comments:

Reviewer #1 (Remarks to the Author):

In this revision the authors have performed several experiments to try to address my previous concerns about using transgenic lines that poorly accumulate protein relative to WT controls. The use of complementing mutants K487A and K387A helps, but it would be much nicer if these were WT controls! Perhaps this can be left up to the reader to critique. For me, it limits how excited I can be about the paper.

1) The authors have compared AvrRps4 responses and attempted to control for the lack of accumulation of cDNA R493A lines. Are these results, and the conclusion that the R493A lines are "fine", now transferrable to the Hpa assays? To me, you have to be concerned and retest the relevance of accumulation in each assay, and this is why using lines that don't accumulate protein is a non-starter. Looking at the new Figure S3C, the cDNA line for R493A#2 also doesn't make RNA/is not transcriptionally induced by AvrRps4?

2) Are there Westerns for the accumulation of the transgenic lines for R493K and R493E from Figure 5? Do the lines for the LOF R493E allele make protein? How do the R493K lines accumulate? If you are going to use mutants (K487A, K387A) to say that the R493A allele is "good enough", why not use these R493 mutants, as they are at least in the same residue?

3) Are you sure that the EDS1 cDNA construct used in the mutants is exactly the cDNA construct used for the EDS1 cDNA mutants? Typically, you would expect independent transgenics to express a linear range of protein accumulation from none to WT levels. You should have been able to find lines for WT that match the mutants, and vice-versa, if the mutations don't have an intrinsic effect on protein accumulation. Instead, both functional and non-functional mutants all seem to express the same maximum protein. The previously WT cDNA line doesn't appear to be an unusual hyper-accumulator given the transcript accumulation in S3C.

4) In the new (I think), Figure S3, some of the statistical letter groupings seem incorrect? Looks like the "d"s for gR493A and/or R493As should be "b"s?

Reviewer #3 (Remarks to the Author):

In my opinion the points raised have been addressed sufficiently in the revised manuscript and response letter and the manuscript appears ready for publication.

Response to Reviewer #1

In this revision the authors have performed several experiments to try to address my previous concerns about using transgenic lines that poorly accumulate protein relative to WT controls. The use of complementing mutants K487A and K387A helps, but it would be much nicer if these were WT controls! Perhaps this can be left up to the reader to critique. For me, it limits how excited I can be about the paper.

1) The authors have compared AvrRps4 responses and attempted to control for the lack of accumulation of cDNA R493A lines. Are these results, and the conclusion that the R493A lines are “fine”, now transferrable to the Hpa assays?

*- Thanks for carefully assessing ms R1. We have edited the text to ensure that claims on TNL immunity to Pst AvrRps4 vs Hpa are clear and justified. We disagree that analysis of the cEDS1 K478A and K387A variants is less informative than using WT cEDS1 to test critically our claim that low EDS1 protein accumulation is sufficient for TNL (RRS1S RPS4) resistance signalling. We purposefully chose these mutants as controls because the amino acids are within (K478) and outside (K387) the EDS1 EP domain cavity and have similar or lower accumulation compared to the cR493A lines (Fig. S2C), but are fully functional in TNL immunity to Pst AvrRps4 (Fig. S2D) and Hpa EMWA1 (Fig. S1C and new S2E). We provide below for the reviewer in **Rev-Fig.1 panels A and B** Western blot data for multiple, independent T1 transgenic lines of cEDS1 variants R493A, K478A and K387A and their corresponding Resistant (R) or susceptible (S) phenotypes in TNL (RPP4) ETI to Hpa EMWA1. These assays show the range of EDS1 protein accumulation in independent transgenics and that the TNL resistance/susceptibility scores are consistent between lines of each variant, regardless of basal protein levels. We are not planning to include these data in the ms unless the Reviewer/Editor insist they go in.*

In ms R2 we add panel E to Fig. S2 showing R and S phenotypes (by trypan blue staining) of cK478A and cK387A variant lines #1 and #2 tested alongside cR493A #1 and #2 and cEDS1. These phenotypes agree with our initial R/S scoring of the T1 transgenics (Fig. S1C). Therefore, for these cEDS1 variants, function or non-function in TNL immunity to Hpa is consistent between lines with different protein accumulation. This actually agrees with a study by Stuttmann et al (PLoS Genet. 2016) in which we found that very low amounts of nuclear-enriched EDS1 (less than native EDS1 in Col-0) are sufficient for RPP4 Hpa and TNL RRS1/RPS4 bacterial immunity.

We add a paragraph in Discussion (ln 489-498) summing up evidence that R493A signalling defect is not explained by protein accumulation (see also points below). We have corrected a mistake in legend to Fig. S2 C which stated ‘mock and Pst AvrRps4-treated tissues’. The data are for uninoculated tissues at lower and higher exposure of the same blot.

From the T1 and T3 Hpa resistance data, we argue in the ms that amino acids in EDS1 EP domain are crucial also for TNL immunity signalling to Hpa. We don't know and don't make claims on

whether the timely mobilization of defense gene expression is also needed for TNL *Hpa* resistance. We've added a sentence in Discussion to emphasize this point (ln 537-539). The fact that RPP4 resistance is not recovered in a *cR493A eds1-2 myc2-3* mutant whereas it is recovered to *Pst AvrRps4* (Fig. 3B and C) suggests that qualitatively different transcriptionally mobilized pathways are responsible for resistance. This is an interesting question we would like to interrogate in the future using the mutants described here, and which would require extensive genetic, phenotypic and gene expression analysis.

Please note: the T1 seedling TNL resistance assay was initially described by Stuttmann et al (2011) as a first-pass phenotyping of *Arabidopsis* transgenic material and was shown in Wagner et al (2013 cited) to be a reliable indicator of TNL resistance phenotypes in selected homozygous transgenic lines. As shown in **Rev-Fig.1 panels A and B**, individual BASTA-resistant T1 seedlings are selected and replanted with controls, left to recover and a small amount of leaf tissue taken for Western blot analysis, and then sprayed with TNL-recognized *Hpa*. After scoring R/S, seedlings are rescued for T2 seed. Not all T1 individuals produce 3:1 segregating material. Also, rescue of susceptible seedlings is not always successful. We mark in blots A and B the lines we took on for detailed analysis of single insertion homozygous material in the ms. Seedlings scored as S in panel A and B were highly susceptible because they produced numerous *Hpa* sporangiophores (as monitored under a binocular microscope). The T1 lines did not exhibit intermediate resistance (eg. trailing necrosis), fitting with the subsequent *Hpa* phenotyping of *cR493A* homozygous lines (R1 Fig.s 1B, E and 3C and new Fig. S2E).

Reviewer Figure 1: T1 protein levels of EDS1-EP domain mutants. **A-** cR493A; **B-** cK487A and cK387A. Two-week-old seedlings were sprayed with BASTA, T1 transformants were transferred to fresh pots and protein was sampled. T1 transformants were sprayed with Hpa EMWA1 and disease symptoms scored at 6 dpi based on appearance of sporophores. Resistant mutants with no visible sporulation are marked as **R** and susceptible plants with >5spores/leaf are marked as **S**.

The same T1 Hpa resistance/Western blot assay was done on independent cR493K and cR493E transgenic lines (**Rev-Fig.2 A and B**) shown below. In this selection, because we were losing Hpa-infected plants (we had trouble with necrotrophic fungi which consumed seedlings after Hpa incubation), we selected BASTA^R seedlings from the same batches but in separate pools from protein-tested/Hpa-infected seedlings shown. Therefore, the T1 lines here are not direct progenitors of the homozygous cR493K and cR493E lines #1 and #2 in the ms.

Reviewer Figure 2: T1 protein levels of (A) cR493K and (B) cR493E. Same treatment as in Review figure 1.

To be absolutely sure we are not missing intermediate TNL resistance or susceptibility phenotypes in Hpa infected leaves, we performed new Hpa Emwa1 (RPP4) infection assays on the cR493A

and gR493A lines #1 and #2 compared to the cEDS1, gEDS1 controls and eds1-2 shown in ms R1 Fig. S3. Trypan blue staining data for RPP4 responses are shown in ms R2 new Fig. S3D. These tally with the Pst AvrRps4 R/S p henotypes (Fig. S3B). So, despite gR493A having similar high protein as resistant cEDS1 it is as susceptible to Pst AvrRps4 and Hpa Emwa1 as cR493A lines, supporting our conclusions in ms R1.

2) To me, you have to be concerned and retest the relevance of accumulation in each assay, and this is why using lines that don't accumulate protein is a non-starter. Looking at the new Figure S3C, the cDNA line for R493A#2 also doesn't make RNA/is not transcriptionally induced by AvrRps4?

- Yes cR493A #2 has a low starting level and negligible increase in protein at 24 hpi and mRNA at 8 hpi compared to cR493A #1 in response to Pst AvrRps4 (ms R1 Fig. S3). In two further independent experiments, we retested cR493A lines #1 and #2 basal and Pst AvrRps4-induced protein accumulation profiles and corresponding Pst AvrRps4 bacterial growth. These are shown in **Rev-Fig.3A and B** (- together with cR493K and cR493E lines #1 and #2 - see response below). **We add panel A and B data to ms R2 as Fig. S6A and S6B.** The cR493A lines #1 and #2 behave in the same way in EDS1-controlled outputs to Pst AvrRps4 and Pst ΔCOR AvrRps4 as measured by bacterial growth (Fig. 3A and S3B), expression of defense genes (Fig. S3C) and SA/PR1 accumulation profiles (Fig. 2A, B) based on independent experiments.

Reviewer Figure3: Basal (mock) and Pst AvrRps4 treated protein accumulation profiles of two independent cR493A, cR493K and cR493E transgenic lines. Panels A and B are from independent experiments.

We conclude that the difference between cR493A #1 and #2 protein accumulation (basal and post-infection) does not explain their weak functions. This supports our claim in ms R1 that cR493A signalling defects determined here are not due to low expression or Pst AvrRps4 induced accumulation. There doesn't seem to be a simple relationship between up-regulation of cR493A

variant protein (at 24 hpi) and its weak function in TNL immunity to *Pst AvrRps4*, maybe because there is weak/variable transcriptional feedback.

2) Are there Westerns for the accumulation of the transgenic lines for R493K and R493E from Figure 5? Do the lines for the LOF R493E allele make protein? How do the R493K lines accumulate? If you are going to use mutants (K487A, K387A) to say that the R493A allele is “good enough”, why not use these R493 mutants, as they are at least in the same residue?

- Thanks for highlighting - we include these data in new Fig. S6 A and B (see Rev-Fig.3) and Results text in 369-374. In new Fig. S6 A, B protein accumulation and *Pst AvrRps4* growth data in two independent experiments for cEDS1, cR493A, cR493K and cR493E transgenic lines. We find that mock and *Pst AvrRps4*-induced cR493K protein accumulation differs between transgenic lines #1 and #2, even though they exhibit complete TNL resistance to *Pst AvrRps4* (see ms R1 Fig. 5B). Therefore, lower cR493K accumulation in line #1 appears to be sufficient for EDS1 signalling. cR493K #1 has a similar EDS1 protein accumulation profile as signalling defective cR493A line #1, again consistent with the cR493A line #1 defect in *Pst AvrRps4* immunity not being due to its lower protein basal or induced accumulation. cR493E lines #1 and #2 both have very low basal EDS1 levels (mock). Some cR493E protein accumulates in *Pst AvrRps4* infected tissues, although cR493E line #1 doesn't accumulate mRNA at 8 hpi (Fig. S6C). We feel these experiments critically test and further support our chief conclusion that the R493A mutation in the EDS1 EP domain creates a weak and thus highly informative EDS1 signalling allele.

In Discussion we've added: The homozygous R493A transgenic lines examined here accumulated less EDS1-YFP protein than a control cEDS1 line (Fig. S2C and S3A). We established that R493A lower accumulation does not account for its defect in TNL immunity because EP-domain K487A and K387A mutant lines, as well as R493K line #1, with similar EDS1 protein levels as R493A, expressed full TNL immunity to *Pst AvrRps4* and *Hpa* EMWA1 (Fig. 5A, B and S2D, E, S6A, B). Additionally, EDS1^{R493A} expressed from a genomic construct in gR493A lines was as abundant as cEDS1 and yet fully defective in TNL ETI (Fig. S3). We cannot exclude that very low accumulation of EDS1^{R493E} protein with a negative charge at residue R493 (Fig. S6A, B) is below the threshold needed for a working EDS1 pool, although in a previous study very low amounts of nuclear-enriched EDS1-YFP protein were found to be sufficient for TNL ETI against *Pst AvrRps4* and *Hpa* EMWA1³⁶.

3) Are you sure that the EDS1 cDNA construct used in the mutants is exactly the cDNA construct used for the EDS1 cDNA mutants? Typically, you would expect independent transgenics to express a linear range of protein accumulation from none to WT levels. You should have been able to find lines for WT that match the mutants, and vice-versa, if the mutations don't have an intrinsic effect on protein accumulation. Instead, both functional and non-functional mutants all seem to express the same maximum protein. The previously WT cDNA line doesn't appear to be an unusual hyper-accumulator given the transcript accumulation in S3C.

- Yes, the same vector backbone was used & checked for wild-type and all *cEDS1* variants and in **Rev-Figs. 1 and 2** there is a range of protein levels in T1 transgenic lines. We don't claim that the R493A mutations don't have an effect on steady state *EDS1* protein accumulation. Indeed, we make the point that they and the functional K478A and K387A (ms R1 Fig. S2C), as well as the R493E (**new Fig. S6A, B**) proteins in selected and characterized transgenic lines have generally lower steady state levels than *cEDS1* (please see above points).

4) In the new (I think), Figure S3, some of the statistical letter groupings seem incorrect? Looks like the "d"s for gR493A and/or R493As should be "b"s?

- Thanks for noticing this mislabeling. We've corrected.